# Hidden Parameter Recurrent State Space Models For Changing Dynamics Scenarios

**Vaisakh Shaj**[1,2], **Dieter Büchler**[3], **Rohit Sonker**[4], **Philipp Becker**[1], and **Gerhard Neumann**[1]

[1]Autonomous Learning Robots, KIT, Germany
[2]LCAS, University Of Lincoln, UK
[3]Max Planck Institute for Intelligent Systems, Tübingen, Germany
[4]Indian Institute Of Technology, Kanpur

## Abstract

Recurrent State-space models (RSSMs) are highly expressive models for learning patterns in time series data and system identification. However, these models assume that the dynamics are fixed and unchanging, which is rarely the case in real-world scenarios. Many control applications often exhibit tasks with similar but not identical dynamics which can be modeled as a latent variable. We introduce the Hidden Parameter Recurrent State Space Models (HiP-RSSMs), a framework that parametrizes a family of related dynamical systems with a low-dimensional set of latent factors. We present a simple and effective way of learning and performing inference over this Gaussian graphical model that avoids approximations like variational inference. We show that HiP-RSSMs outperforms RSSMs and competing multi-task models on several challenging robotic benchmarks both on real-world systems and simulations.

## 1 Introduction

System identification, i.e., learning models of dynamical systems from observed data ( Ljung (1998); Gevers (2005)), is a key ingredient of model-predictive control (Camacho & Alba (2013)) and model-based reinforcement learning (RL). State space models (Hamilton (1994); Jordan (2004); Schön et al. (2011)) (SSMs) provide a principled framework for modelling dynamics. Recently there have been several works that fused SSMs with deep (recurrent) neural networks achieving superior results in time series modelling (Haarnoja et al., 2016; Karl et al., 2016) and system identification tasks (Becker et al., 2019; Hafner et al., 2019; Shaj et al., 2020). Learning the dynamics in an encoded latent space allows us to work with high dimensional observations like images and was proven to be successful for planning from high dimensional sensory inputs. However, most of these tasks assume a fixed, unchanging dynamics, which is quite restrictive in real-world scenarios.

Several control applications involve situations where an agent must learn to solve tasks with similar, but not identical, dynamics. A robot playing table tennis may encounter several bats with different weights or lengths, while an agent manipulating a bottle may face bottles with varying amounts of liquid. An agent driving a car may encounter many different cars, each with unique handling characteristics. Humanoids learning to walk may face different terrains with varying slopes or friction coefficients. Any real-world dynamical system might change over time due to multiple reasons, some of which might not be directly observable or understood. For example, in soft robotics small variations in temperature or changes in friction coefficients of the cable drives of a robot can significantly change the dynamics. Similarly, a robot may undergo wear and tear over time which can change its dynamics. Thus, assuming a global model that is accurate throughout the entire state space or duration of use is a limiting factor for using such models in real-world applications.

We found that existing literature on recurrent models fails to model the dynamics accurately in these situations. Thus, we introduce hidden parameter state-space models (HiP-SSM), which allow capturing the variation in the dynamics of different instances through a set of hidden task parameters.

---

Correspondence to Vaisakh Shaj <v.shaj@kit.edu>

We formalize the HiP-SSM and show how to perform inference in this graphical model. Under Gaussian assumptions, we obtain a probabilistically principled yet scalable approach. We name the resulting probabilistic recurrent neural network as Hidden Parameter Recurrent State Space Model (HiP-RSSM). HiP-RSSM achieves state-of-the-art performance for several systems whose dynamics change over time. Interestingly, we also observe that HiP-RSSM often exceeds traditional RSSMs in performance for dynamical systems previously assumed to have unchanging global dynamics due to the identification of unobserved causal effects in the dynamics.

## 2   HIDDEN PARAMETER RECURRENT STATE SPACE MODELS

State space models are Bayesian probabilistic graphical models (Koller & Friedman, 2009; Jordan, 2004) that are popular for learning patterns and predicting behavior in sequential data and dynamical systems. Let $f(\cdot)$ be any arbitrary dynamic model, and let $h(\cdot)$ be any arbitrary observation model. Then, the dynamic systems of a Gaussian state space model can be represented using the following equations

$$\boldsymbol{z}_t = f(\boldsymbol{z}_{t-1}, \boldsymbol{a}_t) + \boldsymbol{u}_t, \quad \boldsymbol{u}_t \sim \mathcal{N}(\boldsymbol{0}, \boldsymbol{Q})$$
$$\boldsymbol{o}_t = h(\boldsymbol{z}_t) + \boldsymbol{v}_t, \quad \boldsymbol{v}_t \sim \mathcal{N}(\boldsymbol{0}, \boldsymbol{R}).$$

Here $\boldsymbol{z}_t$, $\boldsymbol{a}_t$ and $\boldsymbol{o}_t$ are the latent states, control inputs/actions and observations at time t. The vectors $\boldsymbol{u}_t \sim \mathcal{N}(\boldsymbol{0}, \boldsymbol{Q})$ and $\boldsymbol{v}_t \sim \mathcal{N}(\boldsymbol{0}, \boldsymbol{R})$ denote zero-mean Gaussian noise. Further, based on the assumptions made for $f(\cdot)$ and $h(\cdot)$, we can have different variants of Gaussian state space models (Ribeiro, 2004; Wan & Van Der Merwe, 2000). In the linear case, inference can be performed efficiently using Kalman Filters.

Our goal is to learn a state space model that can model the dynamics of partially observable robotic systems under scenarios where the dynamics changes over time. Often, the dynamics of real systems may differ in significant ways from the system our models were trained on. However, it is unreasonable to train a model across all possible conditions an agent may encounter. Instead, we propose a state space model that learns to account for the causal factors of variation observed across tasks at training time, and then infer at test time the model that best describe the system. Thus, we introduce a new formulation namely Hidden Parameter State Space Models (HiP-SSMs), a framework for modeling families of SSMs with different but related dynamics using low-dimensional latent task embeddings. In this section, we formally define the HiP-SSM family in Section 3.1, propose a probabilistically principled inference scheme for HiP-SSMs based on the forward algorithm (Jordan, 2004) in Section 3.2, and finally provide training scheme for the setting in Section 3.3.

### 2.1   HIDDEN PARAMETER STATE SPACE MODELS (HIP-SSMs)

We denote a set of SSMs with transition dynamics $f_{\boldsymbol{l}}$ that are fully described by hidden parameters $\boldsymbol{l}$ and observation model $h$ as a Hidden Parameter SSM (HiP-SSM). In this definition we assume the observation model $h$ to be independent of the hidden parameter $\boldsymbol{l}$ as we only focus on cases where the dynamics changes. HiP-SSMs allows us to extend SSMs to multi-task settings where dynamics can vary across tasks. Defining the changes in dynamics by a latent variable unifies dynamics across tasks as a single global function. In dynamical systems, for example, parameters can be physical quantities like gravity, friction of a surface, or the strength of a robot actuator. Their effects influence the dynamics but not directly observed; and hence $\boldsymbol{l}$ is not part of the observation space and is treated as latent task parameter vector.

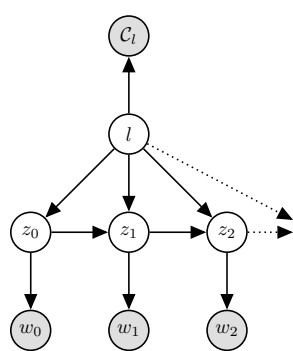

**Figure 1:** The graphical model for a particular instance for the HiP-SSM. The transition dynamics between latent states is a function of the previous latent state, and a specific latent task parameter $\boldsymbol{l}$ which is inferred from a context set of observed past observations. Actions are omitted for brevity.

Formally, a HiP-SSM is defined by a tuple $\{\mathcal{Z}, \mathcal{A}, \mathcal{W}, \mathcal{L}, f, h\}$, where $\mathcal{Z}$, $\mathcal{A}$ and $\mathcal{W}$ are the sets of hidden states $\boldsymbol{z}$, actions $\boldsymbol{a}$, and observations $\boldsymbol{w}$ respectively. $\mathcal{L}$ is the set of all possible latent task parameters and let $p_0(\boldsymbol{l})$ be the prior over these parameters. Thus, a HiP-RSSM describes a class of dynamics and a particular instance of that class is obtained by fixing the parameter vector $\boldsymbol{l} \in \mathcal{L}$. The dynamics $f$ for each instance depend on the value of the hidden parameters $\boldsymbol{l}$.

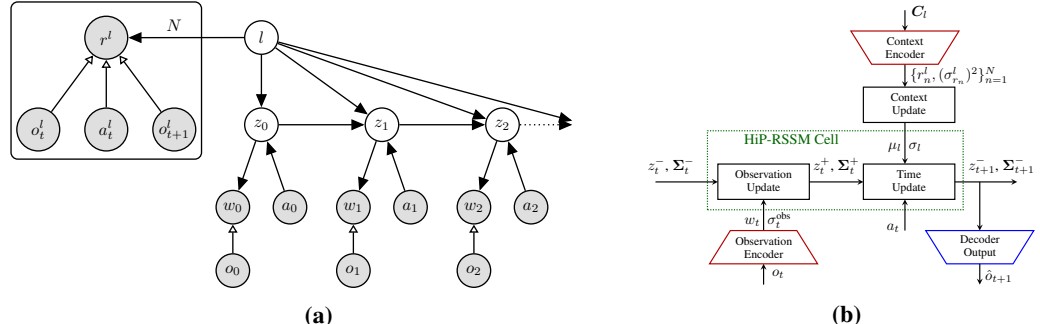

**(a)**                                   **(b)**

**Figure 2: (a)** The Gaussian graphical model corresponding to a particular instance $l \in \mathcal{L}$ for the HiP-RSSM. The posterior of the latent task/context variable is inferred via a Bayesian aggregation procedure described in 2.2.1 based on a set of N interaction histories. The prior over the latent states $z_t^-$ is inferred via task conditional Kalman time update described in 2.2.2 and the posterior over the latent states $z_t^+$ is inferred via Kalman measurement update described in 2.2.3. Here, the hollow arrows denote deterministic transformation leading to implicit distributions, using the context and observation encoders. **(b)** Depicts the schematic Of HiP-RSSM. The output of the task conditional 'Time Update' stage, which forms the prior for the next time step $(z_{t+1}^-, \Sigma_{t+1}^-)$ is decoded to get the prediction of the next observation.

Each instance of a HiP-SSM is an SSM conditioned on $l$. We also make the additional assumption that the parameter vector $l$ is fixed for the duration of the task (i.e. a local segment of a trajectory), and thus the latent task parameter has no dynamics. This assumption considerably simplifies the procedure for inferring the hidden parametrization and is reasonable since dynamics can be assumed to be locally-consistent over small trajectories in most applications (Nagabandi et al., 2018a).

The definition is inspired from related literature on HiP-MDPs (Doshi-Velez & Konidaris, 2016), where the only unobserved variable is the latent task variable. One can connect HiP-SSMs with HiP-MDPs by including rewards to the definition and formalize HiP-POMDPs. However this is left for future work.

## 2.2   LEARNING AND INFERENCE

We perform inference and learning in the HiP-SSM borrowing concepts from both deep learning and graphical model communities following recent works on recurrent neural network models (Haarnoja et al., 2016; Becker et al., 2019), where the architecture of the network is informed by the structure of the probabilistic state estimator. We denote the resulting probabilistic recurrent neural network architecture as Hidden Parameter Recurrent State Space Model (HiP-RSSM).

The structure of the Bayesian network shown in Figure 1 allows tractable inference of latent variables by the forward algorithm (Jordan, 2004; Koller & Friedman, 2009). Since we are dealing with continuous dynamical systems, we assume a Gaussian multivariate distribution over all variables (both observed and hidden) for the graph shown in Figure 1. This assumption has several advantages. Firstly, it makes the inference very similar to the well studied Kalman Filtering approaches. Secondly, the Gaussian assumptions and conditional in-dependencies allows us to have a closed form solution to each of these update procedures which are fully differentiable and can be backpropagated to the deep encoders. The belief update over the hidden variables $z_t$ and $l$ happens in three stages. Similar to Kalman filtering approaches, we have two recursive belief state update stages, the time update and observation update which calculate the prior and posterior belief over the latent states respectively. However, we have an additional hierarchical latent variable $l$ which models the (unobserved) causal factors of variation in dynamics in order to achieve efficient generalization. Hence, we have a third belief update stage to calculate the posterior over the latent task variable based on the observed context set. Each of these three stages are detailed in the sections below:

### 2.2.1   INFERRING THE LATENT TASK VARIABLE (CONTEXT UPDATE)

In this stage, we infer the posterior over the Gaussian latent task variable $l$ based on an observed context set $\mathcal{C}_l$. For any HiP-RSSM instance defined over a target sequence $\mathcal{T} =$

$(\boldsymbol{o}_t, \boldsymbol{a}_t, \boldsymbol{o}_{t+1}, ..., \boldsymbol{o}_{t+N}, \boldsymbol{a}_{t+N})$, over which we intend to perform state estimation/prediction, we maintain a fixed context set $\mathcal{C}_l$. $\mathcal{C}_l$ in our HiP-SSM formalism can be obtained as per the algorithm designer's choice. We choose a $\mathcal{C}_l$ consisting a set of tuples $\mathcal{C}_l = \{\boldsymbol{o}_{t-n}^l, \boldsymbol{a}_{t-n}^l, \boldsymbol{o}_{t-n+1}^l\}_{n=1}^N$. Here each set element is a tuple consisting of the current state/observation, current action and next state/observation for the previous N time steps.

Inferring a latent task variable $l \in \mathcal{L}$ based on an observed context set $\mathcal{C}_l = \{\boldsymbol{x}_n^l\}_{n=1}^N$ has been explored previously by different neural process architectures (Gordon et al., 2018; Garnelo et al., 2018). Neural processes are multi-task latent variable models that rely on deep set functions (Zaheer et al., 2017) to output a latent representation out of varying number of context points in a permutation invariant manner. Similar to Volpp et al. (2020) we formulate context data aggregation as a Bayesian inference problem, where the information contained in $\mathcal{C}_l$ is directly aggregated into the statistical description of $l$ based on a factorized Gaussian observation model of the form $p(\boldsymbol{r}_n^l|l)$, where

$$\boldsymbol{r}_n^l = \mathrm{enc}_{\boldsymbol{r}}(\boldsymbol{o}_{t-n}^l, \boldsymbol{a}_{t-n}^l, \boldsymbol{o}_{t-n+1}^l),$$
$$\left(\boldsymbol{\sigma}_n^l\right)^2 = \mathrm{enc}_{\boldsymbol{\sigma}}(\boldsymbol{o}_{t-n}^l, \boldsymbol{a}_{t-n}^l, \boldsymbol{o}_{t-n+1}^l).$$

Here $n$ is the index of an element from context set $\mathcal{C}_l$. Given a prior $p_0(l) = \mathcal{N}(l|\boldsymbol{\mu}_0, \mathrm{diag}(\boldsymbol{\sigma}_0^2))$ we can compute the posterior $p(l|\mathcal{C}_l)$ using Bayes rule. The Gaussian assumption allows us to get a closed form solution for the posterior estimate of the latent task variable, $p(l|\mathcal{C}_l)$ based on Gaussian conditioning. The factorization assumption further simplifies this update rule by avoiding computationally expensive matrix inversions into a simpler update rule as

$$\boldsymbol{\sigma}_l^2 = \left( (\boldsymbol{\sigma}_0^2)^\ominus + \sum_{n=1}^N \left( (\boldsymbol{\sigma}_n^l)^2 \right)^\ominus \right)^\ominus,$$

$$\boldsymbol{\mu}_l = \boldsymbol{\mu}_0 + \boldsymbol{\sigma}_l^2 \odot \sum_{n=1}^N \left( \boldsymbol{r}_n^l - \boldsymbol{\mu}_0 \right)^2 \oslash \left( \boldsymbol{\sigma}_n^l \right)^2$$

where $\ominus$, $\odot$ and $\oslash$ denote element-wise inversion, product, and division, respectively. Intuitively the mean of the latent task variable $\boldsymbol{\mu}_l$ is a weighted sum of the individual latent observations $\boldsymbol{r}_n^l$, while the variance of the latent task variable $\boldsymbol{\sigma}_l^2$ gives the uncertainty of this task representation.

### 2.2.2 Inferring Prior Latent States over the Next Time Step (Time Update)

The goal of this step is to compute the prior marginal

$$p(\boldsymbol{z}_t|\boldsymbol{w}_{1:t-1}, \boldsymbol{a}_{1:t}, \mathcal{C}_l) = \iint p(\boldsymbol{z}_t|\boldsymbol{z}_{t-1}, \boldsymbol{a}_t, l)p(\boldsymbol{z}_{t-1}|\boldsymbol{w}_{1:t-1}, \boldsymbol{a}_{1:t-1}, \mathcal{C}_l)p(l|\mathcal{C}_l)d\boldsymbol{z}_{t-1}dl. \quad (1)$$

We assume a dynamical model of the following form: $p(\boldsymbol{z}_t|\boldsymbol{z}_{t-1}, \boldsymbol{a}_t, l) = \mathcal{N}(\boldsymbol{A}_{t-1}\boldsymbol{z}_{t-1} + \boldsymbol{B}\boldsymbol{a}_t + \boldsymbol{C}l, \boldsymbol{\Sigma}_{\mathrm{trans}})$ and denote the posterior from the previous time-step by $p(\boldsymbol{z}_{t-1}|\boldsymbol{w}_{1:t-1}, \boldsymbol{a}_{1:t-1}, \mathcal{C}_l) = \mathcal{N}(\boldsymbol{z}_{t-1}^+, \boldsymbol{\Sigma}_{t-1}^+)$. Following Shaj et al. (2020) we assume the action $\boldsymbol{a}_t$ is known and not subject to noise.

At any time t, the posterior over the belief state $\boldsymbol{z}_{t-1}|\boldsymbol{w}_{1:t-1}, \boldsymbol{a}_{1:t-1}$, posterior over the latent task variable $l|\mathcal{C}_l$ and the action $\boldsymbol{a}_t$ are independent of each other since they form a "common-effect" / v-structure (Koller et al., 2007) with the unobserved variable $\boldsymbol{z}_t$. With these independencies and Gaussian assumptions, according to Gaussian Identity 1, it can be shown that calculating the integral in equation 1 has a closed form solution as follows, $p(\boldsymbol{z}_t|\boldsymbol{w}_{1:t-1}, \boldsymbol{a}_{1:t}, \mathcal{C}_l) = \mathcal{N}(\boldsymbol{z}_t^-, \boldsymbol{\Sigma}_t^-)$, where

$$\boldsymbol{z}_t^- = \boldsymbol{A}_{t-1}\boldsymbol{z}_{t-1} + \boldsymbol{B}\boldsymbol{a}_t + \boldsymbol{C}l, \quad (2)$$
$$\boldsymbol{\Sigma}_t^- = \boldsymbol{A}_{t-1}\boldsymbol{\Sigma}_{t-1}^+\boldsymbol{A}_{t-1}^T + \boldsymbol{C}\boldsymbol{\Sigma}_l\boldsymbol{C}^T + \boldsymbol{\Sigma}_{\mathrm{trans}}. \quad (3)$$

**Gaussian Identity 1.** *If $\boldsymbol{u} \sim \mathcal{N}(\boldsymbol{\mu_u} + \boldsymbol{b}, \boldsymbol{\Sigma_u})$ and $\boldsymbol{v} \sim \mathcal{N}(\boldsymbol{\mu_v}, \boldsymbol{\Sigma_v})$ are normally distributed independent random variables and if the conditional distribution $p(\boldsymbol{y}|\boldsymbol{u}, \boldsymbol{v}) = \mathcal{N}(\boldsymbol{Au} + \boldsymbol{b} + \boldsymbol{Bv}, \boldsymbol{\Sigma})$, then marginal $p(\boldsymbol{y}) = \int p(\boldsymbol{y}|\boldsymbol{u}, \boldsymbol{v})p(\boldsymbol{u})p(\boldsymbol{v})d\boldsymbol{u}d\boldsymbol{v} = \mathcal{N}(\boldsymbol{A}\boldsymbol{\mu_u} + \boldsymbol{b} + \boldsymbol{B}\boldsymbol{\mu_v}, \boldsymbol{A}\boldsymbol{\Sigma}_u\boldsymbol{A}^T + \boldsymbol{B}\boldsymbol{\Sigma_v}\boldsymbol{B}^T + \boldsymbol{\Sigma})$*

We use a locally linear transition model $\boldsymbol{A}_{t-1}$ as in Becker et al. (2019) and a non-linear control model as in Shaj et al. (2020). The local linearization around the posterior mean, can be interpreted

as equivalent to an EKF. For the latent task transformation we can either use a linear, locally-linear or non-linear transformation. More details regarding the latent task transformation model can be found in the appendix F.5. Our ablations (Figure 3c) show that a non-linear feedforward neural network $f(.)$ that outputs mean and variances and interact additively gave the best performance in practice. $f(.)$ transforms the latent task moments $\boldsymbol{\mu_l}$ and $\boldsymbol{\sigma_l}$ directly into the latent space of the state space model via additive interactions. The corresponding time update equations are given below:

$$\boldsymbol{z}_t^- = \boldsymbol{A}_{t-1}\boldsymbol{z}_{t-1}^+ + \boldsymbol{b}(\boldsymbol{a}_t) + \boldsymbol{f}(\boldsymbol{\mu_l}),$$
$$\boldsymbol{\Sigma}_t^- = \boldsymbol{A}_{t-1}\boldsymbol{\Sigma}_{t-1}^+ \boldsymbol{A}_{t-1}^T + \boldsymbol{f}(\boldsymbol{\sigma_l}) + \boldsymbol{\Sigma}_{\text{trans}}.$$

### 2.2.3 INFERRING POSTERIOR LATENT STATES / OBSERVATION UPDATE

The goal of this step is to compute the posterior belief $p(\boldsymbol{z}_t|\boldsymbol{o}_{1:t}, \boldsymbol{a}_{1:t}, \boldsymbol{C_l})$. We first map the observations at each time step $\boldsymbol{o}_t$ to a latent space using an observation encoder (Haarnoja et al., 2016; Becker et al., 2019) which emits latent features $\boldsymbol{w}_t$ along with an uncertainty in those features via a variance vector $\boldsymbol{\sigma}_{obs}^t$. We then compute the posterior belief $p(\boldsymbol{z}_t|\boldsymbol{w}_{1:t}, \boldsymbol{a}_{1:t}, \boldsymbol{C_l})$, based on $p(\boldsymbol{z}_t|\boldsymbol{w}_{1:t-1}, \boldsymbol{a}_{1:t}, \boldsymbol{C_l})$ obtained from the time update, the latent observation $(\boldsymbol{w}_t, \boldsymbol{\sigma}_t^{obs})$ and the observation model $\boldsymbol{H}$. This is exactly the Kalman update step, which has a closed form solution as shown below for a time instant $t$,

Kalman Gain: $\qquad \boldsymbol{Q}_t = \boldsymbol{\Sigma}_t^- \boldsymbol{H}^T \left(\boldsymbol{H}\boldsymbol{\Sigma}_t^- \boldsymbol{H}^T + \boldsymbol{I} \cdot \boldsymbol{\sigma}_t^{\text{obs}}\right)^{-1},$

Posterior Mean: $\qquad \boldsymbol{z}_t^+ = \boldsymbol{z}_t^- + \boldsymbol{Q}_t\left(\boldsymbol{w_t} - \boldsymbol{H}\boldsymbol{z}_t^-\right),$

Posterior Covariance: $\qquad \boldsymbol{\Sigma}_t^+ = (\boldsymbol{I} - \boldsymbol{Q}_t\boldsymbol{H})\boldsymbol{\Sigma}_t^-,$

where $\boldsymbol{I}$ denotes the identity matrix. This update is added as a layer in the computation graph (Haarnoja et al. (2016); Becker et al. (2019)). However, the Kalman update involves computationally expensive matrix inversions of the order of $\mathcal{O}(L^3)$, where $L$ is the dimension of the latent state $\boldsymbol{z}_t$. Thus, in order to make the approach scalable we follow the same factorization assumptions as in Becker et al. (2019). This factorization provides a simple way of reducing the observation update equation to a set of scalar operations, bringing down the computational complexity from $\mathcal{O}(L^3)$ to $\mathcal{O}(L)$. More mathematical details on the simplified update equation can be found in appendix F.7.

**Discussion** From a computational perspective, this a Gaussian conditioning layer, similar to section 2.2.1. Both outputs a posterior distribution over latent variables $\boldsymbol{z}$, given a prior $p(\boldsymbol{z})$ and an observation model $p(\boldsymbol{w}|\boldsymbol{z})$, using Bayes rule: $p(\boldsymbol{z}|\boldsymbol{w}) = p(\boldsymbol{w}|\boldsymbol{z})p(\boldsymbol{z})/p(\boldsymbol{w})$. This has a closed form solution because of Gaussian assumptions, which is coded as a layer in the neural network. The observation model is assumed to have the following structure, $P(\boldsymbol{w}|\boldsymbol{z}) = \mathcal{N}(\boldsymbol{H}\boldsymbol{z}, \boldsymbol{\Sigma}_{obs})$.

### 2.3 TRAINING HIP-RSSM FOR EPISODIC ADAPTATION TO CHANGING DYNAMICS

The latent task variable $l$ models a distribution over functions Garnelo et al. (2018), rather than a single function. In our case these functions are latent dynamics functions. In order to train such a model we use a training procedure that reflects this objective, where we form datasets consisting of timeseries, each with a different latent transition dynamics. Thus, we collect a set of trajectories over which the dynamics changes over time. This can be trajectories where a robot picks up objects of different weights or a robot traversing terrain of different slopes. Now we introduce a multi-task setting with a rather flexible definition of task, where each temporal segment of a trajectory can be considered to be a different "task" and the observed context set based on interaction histories from the past $N$ time steps provides information about the current task setting. This definition allows us to have potentially infinite number of tasks/local dynamical systems and the distribution over these tasks/systems is modelled using a hierarchical latent task variable $l$. The formalism is based on the assumption that over these local temporal segments, the dynamics is unchanging. This local consistency in dynamics holds for most real world scenarios (Nagabandi et al., 2018b;a). Yet, our formalism can model the global changes in dynamics at test time, since we obtain a different instance of the HiP-RSSM for each temporal segment based on the observed context set. We also provide a detailed description of the multi-task dataset creation process in the appendix E and a pictorial illustration in appendix D.

**Batch Training.** Let $T \in \mathcal{R}^{B \times N \times D}$ be the batch of local temporal segments with different dynamics which we intend to model with the HiP-RSSM formalism. Given a target batch $T$, HiP-RSSM

can be trained in a supervised fashion similar to popular recurrent architectures like LSTMs or GRUs. However for each local temporal sequence $t \in T$, over which the dynamics is being modelled, we also input a set of the $N$ previous interactions, which forms the context set $C \in \mathcal{R}^{B \times N \times D}$ for inferring the latent task as explained in Section 2.2.1. Processing the context set $C$ adds minimal additional computational/memory constraints as we use a permutation invariant set encoder. The set encoder allows parallelization in processing the context set as opposed to the recurrent processing of target set.

The learnable parameters in the computation graph includes the locally linear transition model $A_t$, the non-linear control factor $b$, the linear/non-linear latent task transformation model $C$, the transition noise $\Sigma_{\text{trans}}$, along with the observation encoder, context encoder and the output decoder.

**Loss Function.** We optimize the root mean square error (RMSE) between the decoder output and the ground truth states. As in Shaj et al. (2020) we use the differences to next state as our ground truth states, as this results in better performance for dynamics learning especially at higher frequencies. In principle, we could train on the Gaussian log-likelihood instead of the RMSE and hence model the variances. Training on RMSE yields slightly better predictions and allows for a fair comparison with deterministic baselines which use the same loss like feed-forward neural networks, LSTMs and meta-learnng algorithms such as MAML. Thus, we report results with the RMSE loss. A probabilistic formulation for this loss function is provided in appendix B.

Gradients are computed using (truncated) backpropagation through time (BPTT) (Werbos (1990)) and clipped. We optimize the objective using the Adam (Kingma & Ba (2014)) stochastic gradient descent optimizer with default parameters.

# 3    RELATED WORK

**Deep State Space Models.** Classical State-space models (SSMs) are popular due to their tractable inference and interpretable predictions. However, inference and learning in SSMs with high dimensional and large datasets are often not scalable. Recently, there have been several works on deep SSMs that offer tractable inference and scalability to high dimensional and large datasets. Haarnoja et al. (2016); Becker et al. (2019); Shaj et al. (2020) use neural network architectures based on closed-form solutions of the forward inference algorithm on SSMs and achieve the state of the art results in state estimation and dynamics prediction tasks. Krishnan et al. (2017); Karl et al. (2016); Hafner et al. (2019) perform learning and inference on SSMs using variational approximations. However, most of these recurrent state-space models assume that the dynamics is fixed, which is a significant drawback since this is rarely the case in real-world applications such as robotics.

**Recurrent Switching Dynamical Systems.** Linderman et al. (2017); Becker-Ehmck et al. (2019); Dong et al. (2020) tries to address the problem of changing/multi-modal dynamics by incorporating additional discrete switching latent variables. However, these discrete states make learning and inference more involved. Linderman et al. (2017) uses auxiliary variable methods for approximate inference in a multi-stage training process, while Becker-Ehmck et al. (2019); Dong et al. (2020) uses variational approximations and relies on additional regularization/annealing to encourage discrete state transitions. On the other hand Fraccaro et al. (2017) uses "soft" switches, which can be interpreted as a SLDS which interpolate linear regimes continuously rather than using truly discrete states. We take a rather different, simpler formalism for modelling changing dynamics by viewing it as a multi-task learning problem with a continuous hierarchical hidden parameter that model the distribution over these tasks. Further our experiments in appendix C.1 show that our model significantly outperfroms the soft switching baseline (Fraccaro et al., 2017).

**Hidden Parameter MDPs.** Hidden Parameter Markov Decision Process (HiP-MDP) (Doshi-Velez & Konidaris, 2016; Killian et al., 2016) address the setting of learning to solve tasks with similar, but not identical, dynamics. In HiP-MDP formalism, the states are assumed to be fully observed. However, we formalize the partially observable setting where we are interested in modelling the dynamics in a latent space under changing scenarios. The formalism is critical for learning from high dimensional observations and dealing with partial observability and missing data. The formalism can be easily extended to HiP-POMDPs by including rewards into the graphical model 1, for planning and control in the latent space (Hafner et al., 2019; Sekar et al., 2020). However this is left as a future work.

| | No Imputation | 50% Imputation |
|---|---|---|
| HiP-RSSM | **2.30 ± 0.043** | **2.47 ± 0.012** |
| RKN | 3.088 ± 0.046 | 3.223 ± 0.014 |
| LSTM | 3.108 ± 0.041 | 3.630 ± 0.097 |
| GRU | 3.287 ± 0.013 | 3.621 ± 0.047 |
| FFNN | 8.150 ± 0.047 | - |
| NP | 5.526 ± 0.030 | - |
| MAML | 7.314 ± 0.021 | - |

**(a)** Pneumatic RMSE $\left(10^{-3}\right)$

| | No Imputation | 50 % Imputation |
|---|---|---|
| HiP-RSSM | **2.833 ± 0.024** | **2.843 ± 0.024** |
| RKN | 3.392 ± 0.062 | 3.398 ± 0.062 |
| LSTM | 3.503 ± 0.006 | 3.736 ± 0.062 |
| GRU | 3.407 ± 0.02 | 3.642 ± 0.153 |
| FFNN | 3.313 ± 0.018 | - |
| NP | **2.765 ± 0.004** | - |
| MAML | 3.202 ± 0.006 | - |

**(b)** Franka RMSE $\left(10^{-4}\right)$

| | No Imputation | 50% Imputation |
|---|---|---|
| HiP-RSSM | **2.96 ± 0.212** | **6.15 ± 0.327** |
| RKN | 7.17 ± 0.017 | 14.66 ± 0.224 |
| LSTM | 9.14 ± 0.026 | 51.21 ± 0.431 |
| GRU | 9.216 ± 0.073 | 53.14 ± 0.242 |
| FFNN | 8.72 ± 0.021 | - |
| NP | 4.57 ± 0.013 | - |
| MAML | 5.04 ± 0.051 | - |

**(c)** Mobile Robot RMSE $\left(10^{-5}\right)$

**Table 1:** Prediction Error in RMSE for (a) pneumatic muscular arm (4.1) and (b) Franka Arm manipulating varying loads (4.2) for both fully observable and partially observable scenarios.

**Meta Learning For Changing Dynamics.** There exists a family of approaches that attempt online model adaptation to changing dynamics scenarios via meta-learning (Nagabandi et al., 2018a;b). They perform online adaptation on the parameters of the dynamics models through gradient descent (Finn et al., 2017) from interaction histories. Our method fundamentally differs from these approaches in the sense that we do not perform a gradient descent step at every time step during test time, which is computationally impractical in modern robotics, where we often deal with high frequency data. We also empirically show that our approach adapts better especially in scenarios with non-markovian dynamics, a property that is often encountered in real world robots due to stiction, slip, friction, pneumatic/hydraulic/cable-driven actuators etc. Sæmundsson et al. (2018); Achterhold & Stueckler (2021) on the other hand learn context conditioned hierarchical dynamics model for control in a formalism similar to HiP-MDPs. The former meta-learn a context conditioned gaussian process while the later use a context conditioned determisitic GRU. Our method on the other hand focuses on a principled probabilistic formulation and inference scheme for context conditioned recurrent state space models, which is critical for learning under partial observabilty/high dimensional observations and with noisy and missing data.

## 4    EXPERIMENTS

This section evaluates our approach on a diverse set of dynamical systems from the robotics domain in simulations and real systems. We show that HiP-RSSM outperforms contemporary recurrent state-space models (RSSMs) and recurrent neural networks (RNNs) by a significant margin under changing dynamics scenarios. Further, we show that HiP-RSSM outperforms these models even under situations with partial observability/ missing values. We also baseline our HiP-RSSM with contemporary multi-task models and improve performance, particularly in modelling non-Markovian dynamics and under partial observability. Finally, the visualizations of the Gaussian latent task variables in HiP-RSSM demonstrates that they learn meaningful representations of the causal factors of variations in dynamics in an unsupervised fashion.

We consider the following baselines:

- RNNs - We compare our method to two widely used recurrent neural network architectures, LSTMs (Hochreiter & Schmidhuber, 1997) and GRUs (Cho et al., 2014).

- RSSMs - Among several RSSMs from the literature, we chose RKN (Becker et al., 2019) as these have shown excellent performance for dynamics learning (Shaj et al., 2020) and relies on exact inference as in our case.

- Multi Task Models - We also compare with state of the art multi-task learning models like Neural Processes (NPs) and MAML (Nagabandi et al., 2018a). Both models receive the same context information as in HiP-RSSM.

In case of recurrent models we replace the HiP-RSSM cell with a properly tuned LSTM, GRU and RKN Cell respectively, while fixing the encoder and decoder architectures. For the NP baseline we use the same context encoder and aggregation mechanism as in HiP-RSSM to ensure a fair comparison. We create partially observable settings by imputing 50% of the observations during inference. More details regarding the baselines and hyperparameters can be found in the appendix.

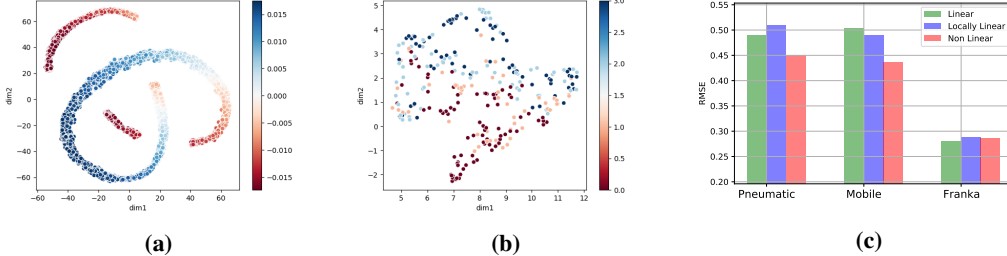

**Figure 3:** (a) and (b) shows the tSNE (Van der Maaten & Hinton, 2008) plots of the latent task embeddings produced from randomly sampled instances of HiP-RSSM for two different robots. (a) The wheeled robot discussed in section 4.3 traversing terrains of varying slopes. The color coding indicates the average gradients of the terrain for each of these instances. These can have either positive or negative values. (b) The Franka manipulator with loads of varying weights attached to the end-effector. The color coding indicated weights ranging from 0 to 3 kgs. (c) An ablation on the performance of different task transformation models discussed in section 2.2.2.

## 4.1 Soft Robot Playing Table Tennis

We first evaluate our model on learning the dynamics of a pneumatically actuated muscular robot. This four Degree of Freedom (DoF) robotic arm is actuated by Pneumatic Artificial Muscles (PAMs) (Büchler et al., 2016). The data consists of trajectories of hitting movements with varying speeds while playing table tennis (Büchler et al., 2020). This robot's fast motions with high accelerations are complicated to model due to hysteresis and hence require recurrent models (Shaj et al., 2020).

We show the prediction accuracy in RMSE in Table 5a. We observe that the HiP-RSSM can outperform the previous state of the art predictions obtained by recurrent models. Based on the domain knowledge, we hypothesise that the latent context variable captures multiple unobserved causal factors of variation that affect the dynamics in the latent space, which are not modelled in contemporary recurrent models. These causal factors could be, in particular, the temperature changes or the friction due to a different path that the Bowden trains take within the robot. Disentangling and interpreting these causal factors can be exciting and improve generalisation, but it is out of scope for the current work. Further, we find that the multi-task models like NPs and MAML fail to model these dynamics accurately compared to all the recurrent baselines because of the non-markovian dynamics resulting from the high accelerations in this pneumatically actuated robot.

## 4.2 Robot Manipulation With Varying Loads

We collected the data from a 7 DoF Franka Emika Panda manipulator carrying six different loads at its end-effector. It involved a mix of movements of different velocities from slow to swift motions covering the entire workspace of the robot. We chose trajectories with four different loads as training set and evaluated the performance on two unseen weights, which results in a scenario where the dynamics change over time. Here, the causal factor of variation in dynamics is the weights attached to the end-effector and assumed to be unobserved.

We show the prediction errors in RMSE in Table 1c. HiP-RSSMs outperforms all recurrent state-space models, including RKN and deterministic RNNs, in modelling these dynamics under fully observable and partially observable conditions. The multi-task baselines of NPs and MAML perform equally well under full observability for this task because of the near Markovian dynamics of Franka Manipulator, which often does not need recurrent models. However, HiP-RSSMs have an additional advantage in that these are naturally suited for partially observable scenarios and can predict ahead in a compact latent space, a critical component for recent success in model-based control and planning (Hafner et al., 2019).

## 4.3 Robot Locomotion In Terrains Of Different Slopes

Wheeled mobile robots are the most common types of robots being used in exploration of unknown terrains where they may face uneven and challenging terrain. We set up an environment using a

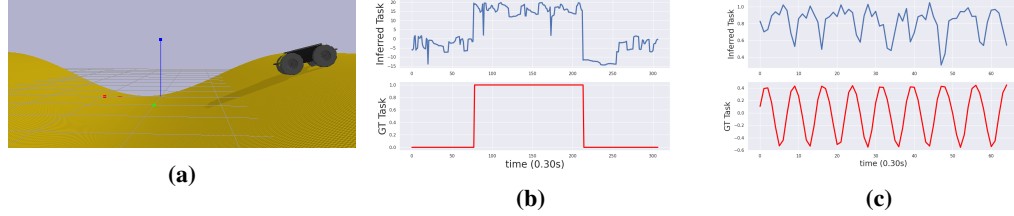

**(a)**            **(b)**            **(c)**

**Figure 4:** (a) Simulated environment of a Wheeled Robot traversing terrains of different slopes. (b) and (c) shows how the one dimensional UMAP (McInnes et al., 2018) embedding of the inferred latent task variable (top blue plots) changes at test time when an agent undergoes changes in its dynamics for the franka robot and mobile robot respectively. An indicator of the Ground Truth (GT) Task (bottom red plots) variables are also given. In case of the Franka Robot, the groundtruth (GT) tasks denotes the switching of dynamics between 0 kg (free motion) and 2.5 kg loads. In case of the mobile robot the groundtruth (GT) tasks denoted slopes of the terrain.

Pybullet simulator (Coumans & Bai, 2016) where a four-wheeled mobile robot traverses an uneven terrain of varying steepness generated by sinusoidal functions (Sonker & Dutta, 2020) as shown in 4a. This problem is challenging due to the highly non-linear dynamics involving wheel-terrain interactions. In addition, the varying steepness levels of the terrain results in a changing dynamics scenario, which further increases the complexity of the task.

We show the prediction errors in RMSE in Table 1c. When most recurrent models, including RSSMs and deterministic RNNs, fail to model these dynamics, HiP-RSSMs are by far the most accurate in modelling these challenging dynamics in the latent space. Further, the HiP-RSSMs perform much better than state of the art multi-task models like NPs and MAML.

We finally visualize the latent task representations using TSNE in Figure 3a. As seen from the plot, those instances of the HiP-SSMs under similar terrain slopes get clustered together in the latent task space, indicating that the model correctly identifies the causal effects in the dynamics in an unsupervised fashion.

### 4.4 VISUALIZING CHANGING HIDDEN PARAMETERS AT TEST TIME OVER TRAJECTORIES WITH VARYING DYNAMICS

We finally perform inference using the trained HiP-RSSM in a multi-task / changing dynamics scenario where the dynamics continuously changes over time. We use the inference procedure described in appendix D based on a fluid definition for "task" as the local dynamics in a temporal segment. We plot the global variation in the latent task variable captured by each instance of the HiP-RSSM over these local temporal segments using the dimensionality reduction technique UMAP (McInnes et al., 2018). As seen in figures 4b and 4c, the latent task variable captures these causal factors of variations in an interpretable manner.

## 5 CONCLUSION

We proposed HiP-RSSM, a probabilistically principled recurrent neural network architecture for modelling changing dynamics scenarios. We start by formalizing a new framework, HiP-SSM, to address the multi-task state-space modelling setting. HiP-SSM assumes a shared latent state and action space across tasks but additionally assumes latent structure in the dynamics. We exploit the structure of the resulting Bayesian network to learn a universal dynamics model with latent parameter $l$ via exact inference and backpropagation through time. The resulting recurrent neural network, namely HiP-RSSM, learns to cluster SSM instances with similar dynamics together in an unsupervised fashion. Our experimental results on various robotic benchmarks show that HiP-RSSMs significantly outperform state of the art recurrent neural network architectures on dynamics modelling tasks. We believe that modelling the dynamics in the latent space which disentangles the state, action and task representations can benefit multiple future applications including planning/control in the latent space and causal factor identification.

## 6 Aknowledgements

We thank Michael Volpp, Onur Celik, Marc Hanheide and the anonymous reviewers for valuable remarks and discussions which greatly improved this paper. The authors acknowledge support by the state of Baden-Württemberg through bwHPC and the Lichtenberg high performance computer of the TU Darmstadt. This work was supported by the EPSRC UK (project NCNR, National Centre for Nuclear Robotics,EP/R02572X/1).

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

# A  PROOF FOR GAUSSIAN IDENTITY 1

This section provides a proof for Gaussian Identity 1. First we derive an expression for the joint distribution $p(\boldsymbol{u}, \boldsymbol{v}, \boldsymbol{y})$.

**Gaussian Identity 2.** *If $\boldsymbol{u} \sim \mathcal{N}(\boldsymbol{\mu}_u + b, \boldsymbol{\Sigma}_u)$ and $v \sim \mathcal{N}(\boldsymbol{\mu}_v, \boldsymbol{\Sigma}_v)$ are normally distributed independent random variables and if conditional distribution $p(\boldsymbol{y}|\boldsymbol{u}, \boldsymbol{v}) = \mathcal{N}(\boldsymbol{A}\boldsymbol{u} + b + \boldsymbol{B}v, \boldsymbol{\Sigma})$, the joint distribution has an expression as follows:*

$$\begin{pmatrix} \mathbf{u} \\ \mathbf{v} \\ \mathbf{y} \end{pmatrix} \sim \mathcal{N}\left( \begin{pmatrix} \boldsymbol{\mu}_u \\ \boldsymbol{\mu}_v \\ \mathbf{A}\mu_u + b + \mathbf{B}\mu_v \end{pmatrix}, \begin{pmatrix} \boldsymbol{\Sigma}_u & 0 & \boldsymbol{\Sigma}_u \mathbf{A}^\top \\ 0 & \boldsymbol{\Sigma}_v & \boldsymbol{\Sigma}_v \mathbf{B}^\top \\ \mathbf{A}\boldsymbol{\Sigma}_u^\top & \mathbf{B}\boldsymbol{\Sigma}_v^\top & \mathbf{A}\boldsymbol{\Sigma}_u \mathbf{A}^\top + \mathbf{B}\boldsymbol{\Sigma}_v \mathbf{B}^\top + \boldsymbol{\Sigma} \end{pmatrix} \right)$$

**Proof for Identity 2**

Let displacement of a variable $\mathbf{u}$ be denoted by $\Delta \mathbf{u} = \mathbf{u} - \langle \mathbf{u} \rangle$.

Since $\mathbf{u}$ and $\mathbf{v}$ are independent, covariances $\langle \Delta \mathbf{u} \Delta \mathbf{v}^\top \rangle = 0$.

We can write $\mathbf{y} = \mathbf{A}\boldsymbol{u} + b + \mathbf{B}\boldsymbol{v} + \boldsymbol{\epsilon}$, where $\boldsymbol{\epsilon} \sim \mathcal{N}(\mathbf{0}, \boldsymbol{\Sigma})$ and $b$ is a constant. Then we have covariance $\langle \Delta \mathbf{u} \Delta \mathbf{y}^\top \rangle = \langle \Delta \mathbf{u}(\mathbf{A}\Delta \mathbf{u} + \mathbf{B}\Delta \mathbf{v} + \Delta \epsilon)^\top \rangle = \langle \Delta \mathbf{u} \Delta \mathbf{u}^\top \rangle \mathbf{A}^\top + \langle \Delta \mathbf{u} \Delta \mathbf{v}^\top \rangle \mathbf{B}^\top + \langle \Delta \mathbf{u} \Delta \epsilon^\top \rangle$. Since $\langle \Delta \mathbf{u} \Delta \mathbf{v}^\top \rangle = \langle \Delta \mathbf{u} \Delta \epsilon^\top \rangle = 0$ we therefore have $\langle \Delta \mathbf{u} \Delta \mathbf{y}^\top \rangle = \boldsymbol{\Sigma}_u \mathbf{A}^\top$.

The derivations for covariances $\langle \Delta \mathbf{v} \Delta \mathbf{y}^\top \rangle$ follows similarly and the corresponding covariance has the expression, $\langle \Delta \mathbf{v} \Delta \mathbf{y}^\top \rangle = \boldsymbol{\Sigma}_v \mathbf{B}^\top$.

Similarly, $\langle \Delta \mathbf{y} \Delta \mathbf{y}^\top \rangle = \langle (\mathbf{A}\Delta \mathbf{u} + \mathbf{B}\Delta \mathbf{v} + \Delta \epsilon)(\mathbf{A}\Delta \mathbf{u} + \mathbf{B}\Delta \mathbf{v} + \Delta \epsilon)^\top \rangle = \mathbf{A} \langle \Delta \mathbf{u} \Delta \mathbf{u}^\top \rangle \mathbf{A}^\top + \mathbf{B} \langle \Delta \mathbf{v} \Delta \mathbf{v}^\top \rangle \mathbf{B}^\top + \langle \Delta \epsilon \Delta \epsilon^\top \rangle = \mathbf{A}\boldsymbol{\Sigma}_u \mathbf{A}^\top + \mathbf{B}\boldsymbol{\Sigma}_v \mathbf{B}^\top + \Sigma$. The result follows.

**Gaussian Identity 3** (Gaussian Marginalization). *If*

$$\begin{pmatrix} \mathbf{u} \\ \mathbf{v} \\ \mathbf{y} \end{pmatrix} \sim \mathcal{N}\left( \begin{pmatrix} \boldsymbol{\mu}_u \\ \boldsymbol{\mu}_v \\ \boldsymbol{\mu}_z \end{pmatrix}, \begin{pmatrix} \boldsymbol{\Sigma}_{uu} & \boldsymbol{\Sigma}_{uv} & \boldsymbol{\Sigma}_{uy} \\ \boldsymbol{\Sigma}_{uv}^\top & \boldsymbol{\Sigma}_{vv} & \boldsymbol{\Sigma}_{vy} \\ \boldsymbol{\Sigma}_{uy}^\top & \boldsymbol{\Sigma}_{vy}^\top & \boldsymbol{\Sigma}_{yy} \end{pmatrix} \right)$$

*then marginal over y is given as $p(\boldsymbol{y}) = \int_{\boldsymbol{u}, \boldsymbol{v}} p(\boldsymbol{y}|\boldsymbol{u}, \boldsymbol{v}) p(\boldsymbol{u}) p(\boldsymbol{v}) d\boldsymbol{u} d\boldsymbol{v} = \mathcal{N}(\boldsymbol{\mu}_y, \boldsymbol{\Sigma}_{yy})$*

**Proof For Identity 3**

We refer to Bishop (2006) for the derivation, which requires calculation of the Schur complement as well as completing the square of the Gaussian p.d.f. to integrate out the variable. The given derivation (Bishop, 2006) for two variable multivariate Gaussians can be extended to 3 variable case WLOG.

**Proof For Identity 1** is immediate from Identity 2 and Identity 3.

# B  PROBABILISTIC FORMULATION OF HIP-RSSM TRAINING OBJECTIVE

The training objective for the HiP-RSSM involves maximizing the posterior predictive log-likelihood given below:

$$\sum_{t=1}^{T} \sum_{b=1}^{N} \log p(\hat{\boldsymbol{o}}_{t+1}^b | \boldsymbol{C}_l, \boldsymbol{o}_{1:t}^b) = \sum_{t=1}^{T} \sum_{b=1}^{N} \log \int p(\hat{\boldsymbol{o}}_{t+1}^b | \boldsymbol{z}_{t+1}^b) p(\boldsymbol{z}_{t+1}^b | \boldsymbol{o}_{1:t}^m, \boldsymbol{C}_l) d\boldsymbol{z}_{t+1}^b. \quad (4)$$

Here, for any time $t$, $\hat{\boldsymbol{o}}_{t+1}$ are the ground truth observations for the next time step and the latent state prior $p(\boldsymbol{z}_{t+1}^b | \boldsymbol{o}_{1:t}^m, \boldsymbol{C}_l)$ as discussed in the Section 2.2.2, have closed form solutions. We approximate the objective in Eq.(4) using a deterministic variational approximation, similar to Becker et al. (2019).

We employ a Gaussian approximation of the posterior predictive log likelihood of the form $p(\hat{\boldsymbol{o}}_{t+1}^b | \boldsymbol{C}_l, \boldsymbol{o}_{1:t}^b) \approx \mathcal{N}(\boldsymbol{\mu}_{\boldsymbol{o}_{t+1}^b}, \text{diag}((\boldsymbol{\sigma}_{\boldsymbol{o}_{t+1}^b})^2))$. Here $\boldsymbol{\mu}_{\boldsymbol{o}_{t+1}^b} = \text{dec}_{\boldsymbol{\mu}}(\boldsymbol{z}_{t+1}^b)$ and $\boldsymbol{\sigma}_{\boldsymbol{o}_t^b} = \text{dec}_{\Sigma}(\boldsymbol{\Sigma}_{t+1}^b))$, where $\text{dec}_{\boldsymbol{\mu}}()$ and $\text{dec}_{\boldsymbol{\Sigma}}()$ denote the parts of the output decoder that are responsible for decoding the latent means and latent variances respectively. This decoder can be interpreted

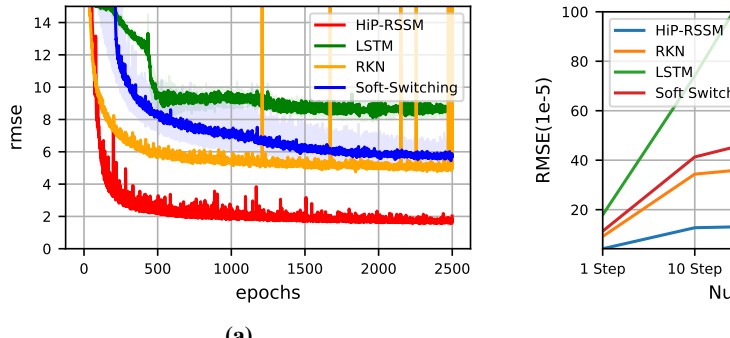

**(a)**          **(b)**

**Figure 5:** Comparison of different algorithms for Wheeled Mobile Robot in terms of (a) moving average of decoder error in normalized RMSE for the test set, plotted against training epochs (b) multi-step ahead prediction error in RMSE.

as a "moment matching network", computing the moments of $\boldsymbol{o}_t^b$ given the moments of $\boldsymbol{z}_t^b$ (Volpp et al., 2020; Becker et al., 2019). This allows to evaluate an approximation to the objective in a deterministic manner.

Alternatively, we could ignore the variances and only focus on the mean predictions by training our model with a RMSE loss. Training on RMSE yields slightly better predictions and allows for a fair comparison with deterministic baselines (Feed Forward NN, LSTM, GRU, MAML etc) and hence we report results with the RMSE loss.

## C  ADDITIONAL EXPERIMENTS

### C.1  COMPARISON TO SOFT SWITCHING BASELINE

We also implement a soft-switching baseline similar to Fraccaro et al. (2017), where a soft mixture of dynamics is implemented in the latent transition dynamics (Kalman time update). Similar to Fraccaro et al. (2017), we now globally learn $K$ constant transition matrices $\boldsymbol{A}^{(k)}$ and control matrices $\boldsymbol{B}^{(k)}$. An interpolation is done between these using a "dynamics parameter network" (Fraccaro et al., 2017) $\alpha_t = \boldsymbol{\alpha}_t (\mathbf{w}_{0:t-1})$. The dynamics parameter network is implemented with a recurrent neural network with LSTM cells that takes at each time step the mean of the encoded observation $w_t$ as input and recurses $\mathbf{d}_t = \mathrm{LSTM}(\mathbf{w}_{t-1}, \mathbf{d}_{t-1})$ and $\boldsymbol{\alpha}_t = \mathrm{softmax}(\mathbf{d}_t)$. The output of the dynamics parameter network is weights that sum to one, $\sum_{k=1}^K \alpha_t^{(k)}(\mathbf{w}_{0:t-1}) = 1$. These weights choose and interpolate between $K$ different operating modes:

$$\mathbf{A}_t = \sum_{k=1}^K \alpha_t^{(k)}(\mathbf{w}_{0:t-1})\, \mathbf{A}^{(k)}, \quad \mathbf{B}_t = \sum_{k=1}^K \alpha_t^{(k)}(\mathbf{w}_{0:t-1})\, \mathbf{B}^{(k)}$$

Authors interpret the weighted sum as a soft mixture of $K$ different Linear Gaussian SSMs whose time-invariant matrices are combined using the time varying weights $\boldsymbol{\alpha}_t$. In practice, each of the $K$ sets $\{\mathbf{A}^{(k)}, \mathbf{B}^{(k)}\}$ models different/changing dynamics, that will dominate when the corresponding $\alpha_t^{(k)}$ is high.

Figure 5 compares HiP-RSSM with different recurrent architectures including the soft-switching baseline. As seen in figure 5, HiP-RSSM clearly outperforms the soft-switching baseline both in terms of convergence speed and also mult-step ahead predictions. More details regarding the multi-step ahead training procedure can be found in appendix C.3.

### C.2  ABLATION ON CONTEXT ENCODER

In table 2, we report the details of evaluating our bayesian aggregation based context set encoder (discussed in 2.2.1) against the a causal / recurrent encoder that takes into account the temporal structure of the context data. We used a probabilistic recurrent encoder (Becker et al.,

2019), whose mean and variance from the last time step is used to infer the posterior latent task distribution $p(l|\mathcal{C}_l) = \mathcal{N}(\boldsymbol{\mu_l}, \text{diag}(\boldsymbol{\sigma_l^2}))$. The dimensions of latent task parameters obtained from both the set and recurrent encoders are kept the same (60).

The reported experiments are conducted on data from wheeled mobile robot discussed in section 4.3. As reported in Table 2 the permutation invariant set encoder outperforms the recurrent encoder by a good margin in terms of prediction accuracy for both fully and partially observable scenarios. Additionally the set encoder is far more efficient in terms of computational time required for training as seen from the time taken per epoch for each of these cases.

**Table 2:** Comparison between the permutation invariant set encoder and recurrent encoder. The performance is measured in terms of prediction RMSE (10-5) and mean of the training time per epoch (in seconds) over 5 runs.

|                  | No Imputation RMSE | 50% Imputation RMSE | Training Time Per Epoch |
|------------------|--------------------|---------------------|-------------------------|
| Set Encoder      | **2.96 ± 0.212**   | **6.15 ± 0.327**    | **6.71**                |
| Recurrent Encoder| 5.10 ± 0.041       | 10.12 ± 0.112       | 14.13                   |

### C.3 MULTI-STEP AHEAD PREDICTIONS

In figure 5b, we compare the results of multi-step ahead predictions for upto 50 steps of HiP-RSSM with recurrent baselines like RKN(Becker et al., 2019) and LSTM. Inorder to train the recurrent models for multi step ahead predictions, we removed three-quarters of the observations from the temporal sequence and tasked the models with imputing those missing observations, only based on the knowledge of available actions/control commands, i.e., we train the models to perform action conditional future predictions to impute missing observations. The imputation employs the model for multi-step ahead predictions in a convenient way (Shaj et al., 2020). One could instead also go for a dedicated multi-step loss function as in approaches like Finn et al. (2016).

As seen in figure 5b, HiP-RSSM clearly outperforms contemporary recurrent models for multi-step ahead prediction tasks since it takes into account additional causal factors of variation (slopes of the terrain for this robot) in the latent dynamics in an unsupervised manner.

## D HiP-RSSM DURING TEST TIME / INFERENCE

We perform inference using HiP-RSSM at test time on a trajectory with varying dynamics using algorithm 1. A pictorial representation of the same is given in the figure 6. We use this inference scheme to visualize how the latent variable $l$, that describe different instances of a HiP-RSSM over different temporal segments, evolve at a global level. The visualizations are reported in figures 4b and 4c in the main text.

---

**Algorithm 1:** HiP-RSSM Test Time Inference

---

**Required:** Trained HiP-RSSM Model
**Required:** A time series $\tau$ of length $K >> N$
Divide the time series $\tau$ into non-overlapping windows $T_l$ of length N. Let
  $T = \{T_1, T_2, T_3, ., ., .\}$ be the ordered list of all temporal segments, sorted in the ascending
  order of time of occurrence.
**foreach** *each time window $T_l \in T$* **do**

      1. maintain a context set $C_l$ consisting of N previous interactions;

      2. infer posterior latent task variable $p(l|\mathcal{C}_l)$ using context update stage as in section 2.2.1;

      3. using the posterior over latent task variable $l|\mathcal{C}_l$ and observations in sequence $T_l$ to perform sequential Bayesian inference over the state space model using Kalman observation update (2.2.3) and task conditional Kalman time update; (2.2.2)

**end**

---

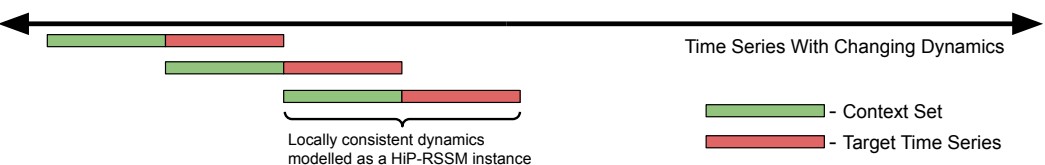

**Figure 6:** HiP-RSSM Inference Under Changing Dynamics Scenarios

## E  MULTI-TASK DATASET CREATION

---

**Algorithm 2:** Multi Task Dataset Creation For Training HiP-RSSM

---

**Required:** A set $S$ of trajectories of changing dynamics

$D \leftarrow \phi$

**foreach** *each trajectory $\tau \in S$* **do**

    1. divide the trajectory $\tau$ into non-overlapping windows $T_l$ of length N. Let $T = \{T_1, T_2, T_3, ., ., .\}$ be the list of all temporal segments/time-series.

    **foreach** *each time window $T_l \in T$* **do**

        (a) maintain a context set $C_l$ consisting of N previous interactions;

        (b) update $D \leftarrow D \cup \{C_l, T_l\}$

    **end**

**end**

**Output**: Output $D$ consisting of batch of context and target sets.

---

## F  IMPLEMENTATION DETAILS

### F.1  CONTEXT SET ENCODER AND LATENT TASK REPRESENTATION

The HiP-RSSM maps a set of previous interaction histories, $\{o^l_{t,n}, a^l_{t,n}, o^l_{t+1,n}\}^N_{n=1}$, to a set of latent features and an estimate of uncertainty in those features, $\{r^l_n, (\sigma^l_n)^2\}^N_{n=1}$, using a context encoder. We use a feed-forward neural network as the encoder in all of our experiements since we deal with high dimensional vectors as our observations. However, depending upon the nature of observations, we could use different encoder architectures.

The set of latent features and uncertainties, $\{r^l_n, (\sigma^l_n)^2\}^N_{n=1}$, are further aggregated in a probabilistically principled manner using bayesian aggregation operator discussed in 2.2.1 to get a gaussian latent task variable, with a mean ($\mu_l$) and diagonal covariance ($\sigma_l$). Intuitively the context encoder learns to weight the contribution from each observation in the context set based on bayesian priniciples and emits a probabilistic representation of the latent task.

### F.2  OBSERVATION ENCODER AND LATENT STATE REPRESENTATION

HiP-RSSM transforms the observations at each time step $o_t$ to a latent space using an encoder network which emits latent features $w_t$ and an estimate of the uncertainty in those features via a variance vector $\sigma^{\text{obs}}_t$.

The probabilistic recurrent module uses a latent state vector $z_t$ and corresponding covariance $\Sigma_t$ whose transitions are governed by the update rules in the HiP-RSSM cell. The latent state vector $z_t$ has been designed to contain two conceptual parts, a vector $p_t$ for holding information that can directly be extracted from the observations and a vector $m_t$ to store information inferred over time, e.g., velocities. The former is referred to as the observation or upper part and the latter as the memory or lower part of the latent state as in Becker et al. (2019). For an ordinary dynamical system and images as observations the former may correspond to positions while the latter corresponds to velocities. The corresponding posterior and prior covariance matrices $\Sigma^+_t$ and $\Sigma^-_t$ have a chosen factorized representation to yield simple Kalman updates, i.e.,

$$\boldsymbol{\Sigma}_t = \begin{bmatrix} \boldsymbol{\Sigma}_t^{\mathrm{u}} & \boldsymbol{\Sigma}_t^{\mathrm{s}} \\ \boldsymbol{\Sigma}_t^{\mathrm{s}} & \boldsymbol{\Sigma}_t^{\mathrm{l}} \end{bmatrix},$$

where each of $\boldsymbol{\Sigma}_t^{\mathrm{u}}, \boldsymbol{\Sigma}_t^{\mathrm{s}}, \boldsymbol{\Sigma}_t^{\mathrm{l}} \in \mathbb{R}^{m \times m}$ is a diagonal matrix. The vectors $\boldsymbol{\sigma}_t^{\mathrm{u}}, \boldsymbol{\sigma}_t^{\mathrm{l}}$ and $\boldsymbol{\sigma}_t^{\mathrm{s}}$ denote the vectors containing the diagonal values of those matrices. This structure with $\boldsymbol{\Sigma}_t^{\mathrm{s}}$ ensures that the correlation between the memory and the observation parts are not neglected as opposed to the case of designing $\boldsymbol{\Sigma}_t$ as a diagonal covariance matrix. This representation was exploited to avoid the expensive and numerically problematic matrix inversions involved in the KF equations as shown below.

## F.3 LOCALLY LINEAR TRANSITION MODEL

The state transitions in the time update stage (section 2.2.2) of the HiP-RSSM Cell is governed by a locally linear transition model as in Becker et al. (2019). To obtain a locally linear transition model, the HiP-RSSM gloablly learns $K$ constant transition matrices $\boldsymbol{A}^{(k)}$ and interpolate between them using state dependent coefficients $\alpha^{(k)}(\boldsymbol{z}_t^+)$, i.e., $\boldsymbol{A}_t = \sum_{k=0}^{K} \alpha^{(k)}(\boldsymbol{z_t^+})\boldsymbol{A}^{(k)}$, where $\boldsymbol{z}_t^+$ is the mean of the posterior distribution at time step t. So we locally linearize $A_t$ around the posterior mean, which can be interpreted as equivalent to what an EKF do with a Jacobian. A small neural network with softmax output is used to learn $\alpha^{(k)}$.

## F.4 CONTROL MODEL

To achieve action conditioning within the recurrent cell, we include a control model $\boldsymbol{b}(\boldsymbol{a}_t)$ in addition to the locally linear transition model $\boldsymbol{A}_t$. $\boldsymbol{b}(\boldsymbol{a}_t) = \boldsymbol{f}(\boldsymbol{a}_t)$, where $\boldsymbol{f}(.)$ can be any non-linear function approximator. We use a multi-layer neural network regressor with ReLU activations (Shaj et al., 2020).

## F.5 LATENT TASK TRANSFORMATION MODEL

To achieve latent task conditioning within the recurrent cell, we include a task transformation model $(\boldsymbol{c})$, in addition to the locally linear transition model $\boldsymbol{A}_t$ and control model $\boldsymbol{b}$ in the time update stage (section 2.2.2). Though in section 2.2.2, we used the notation for a linear task transformation matrix, $\boldsymbol{C}$, to motivate the additive interaction of latent task variables, $\boldsymbol{\mu}_l$ and $\boldsymbol{\sigma}_l$ in the latent space, the task transformation function can be designed in several ways, i.e.:

(i) **Linear:** $\boldsymbol{c} = \boldsymbol{C}$, where $\boldsymbol{C}$ is a linear transformation matrix. The corresponding time update equations are given below:

$$\boldsymbol{z}_t^- = \boldsymbol{A}_{t-1}\boldsymbol{z}_{t-1}^+ + \boldsymbol{b}(\boldsymbol{a}_t) + \boldsymbol{C}\boldsymbol{\mu_l},$$
$$\boldsymbol{\Sigma}_t^- = \boldsymbol{A}_{t-1}\boldsymbol{\Sigma}_{t-1}^+\boldsymbol{A}_{t-1}^T + \boldsymbol{C}(\boldsymbol{I} \cdot \boldsymbol{\sigma_l})\boldsymbol{C}^T + \boldsymbol{\Sigma}_{\mathrm{trans}}.$$

(ii) **Locally-Linear:** $\boldsymbol{c} = \boldsymbol{C}_t$, where $\boldsymbol{C}_t = \sum_{k=0}^{K} \beta^{(k)}(\boldsymbol{z_t})\boldsymbol{C}^{(k)}$ is a linear combination of k linear control models $\boldsymbol{C}^{(k)}$. A small neural network with softmax output is used to learn $\beta^{(k)}$. The corresponding time update equations are given below:

$$\boldsymbol{z}_t^- = \boldsymbol{A}_{t-1}\boldsymbol{z}_{t-1}^+ + \boldsymbol{b}(\boldsymbol{a}_t) + \boldsymbol{C_t}\boldsymbol{\mu_l},$$
$$\boldsymbol{\Sigma}_t^- = \boldsymbol{A}_{t-1}\boldsymbol{\Sigma}_{t-1}^+\boldsymbol{A}_{t-1}^T + \boldsymbol{C_t}(\boldsymbol{I} \cdot \boldsymbol{\sigma_l})\boldsymbol{C_t}^T + \boldsymbol{\Sigma}_{\mathrm{trans}}.$$

(iii) **Non-Linear:** $\boldsymbol{c} = \boldsymbol{f}$, where $\boldsymbol{f}(.)$ can be any non-linear function approximator. We use a multi-layer neural network regressor with ReLU activations, which transforms the latent task moments $\boldsymbol{\mu}_l$ and $\boldsymbol{\sigma}_l$ directly into the latent space of the state space model via additive interactions. The corresponding time update equations are given below:

$$\boldsymbol{z}_t^- = \boldsymbol{A}_{t-1}\boldsymbol{z}_{t-1}^+ + \boldsymbol{b}(\boldsymbol{a}_t) + \boldsymbol{f}(\boldsymbol{\mu_l}),$$
$$\boldsymbol{\Sigma}_t^- = \boldsymbol{A}_{t-1}\boldsymbol{\Sigma}_{t-1}^+\boldsymbol{A}_{t-1}^T + \boldsymbol{f}(\boldsymbol{\sigma_l}) + \boldsymbol{\Sigma}_{\mathrm{trans}}.$$

In our ablation study (Figure 3c), for the linear and locally linear task transformation models, we assume that the dimension of the latent context variable $l$ and the latent state space $z_t$ are equal. This allows us to work with square matrices which are more convenient. For the non-linear transformation we are free to choose the size of the latent context variable. However for a fairer comparison we keep the dimension of latent task variable to be similar in all three cases. We choose the non-linear task transformation model in HiP-RSSM architecture as this gave the best performance in practice.

### F.6 OBSERVATION MODEL

The latent state space $\mathcal{Z} = \mathbb{R}^n$ of the HiP-RSSM is related to the observation space $\mathcal{W}$ by the linear latent observation model $\boldsymbol{H} = \begin{bmatrix} \boldsymbol{I}_m & \boldsymbol{0}_{m \times (n-m)} \end{bmatrix}$, i.e., $\boldsymbol{w}|\boldsymbol{z} \sim \mathcal{N}(\boldsymbol{H}\boldsymbol{z}, \boldsymbol{\Sigma}_{obs})$ with $\boldsymbol{w} \in \mathcal{W}$ and $\boldsymbol{z} \in \mathcal{Z}$, where $\boldsymbol{I}_m$ denotes the $m \times m$ identity matrix and $\boldsymbol{0}_{m \times (n-m)}$ denotes a $m \times (n-m)$ matrix filled with zeros. Typically, $m$ is set to $n/2$. This corresponds to the assumption that the first half of the state can be directly observed while the second half is unobserved and contains information inferred over time (Becker et al. (2019)).

### F.7 OBSERVATION UPDATE STEP

The observation update discussed in section 2.2.3 involves computing the Kalman gain matrix $\boldsymbol{Q}_t$, which requires computationally expensive matrix inversions that are difficult to backpropagate, at least for high dimensional latent state representations. However, the choice of a locally linear transition model, the factorized covariance $\boldsymbol{\Sigma}_t$, and the special observation model simplify the Kalman update to scalar operations as shown below. As the network is free to choose its own state representation, it finds a representation where such assumptions works well in practice Becker et al. (2019).

Similar to the state, the Kalman gain matrix $\boldsymbol{Q}_t$ is split into an upper $\boldsymbol{Q}_t^{\mathrm{u}}$ and a lower part $\boldsymbol{Q}_t^{\mathrm{l}}$. Both $\boldsymbol{Q}_t^{\mathrm{u}}$ and $\boldsymbol{Q}_t^{\mathrm{l}}$ are squared matrices. Due to the simple latent observation model $\boldsymbol{H} = \begin{bmatrix} \boldsymbol{I}_m & \boldsymbol{0}_{m \times (n-m)} \end{bmatrix}$ and the factorized covariances, all off-diagonal entries of $\boldsymbol{Q}_t^{\mathrm{u}}$ and $\boldsymbol{Q}_t^{\mathrm{l}}$ are zero and one can again work with vectors representing the diagonals, i.e., $\boldsymbol{q}_t^{\mathrm{u}}$ and $\boldsymbol{q}_t^{\mathrm{l}}$. Those are obtained by

$$\boldsymbol{q}_t^{\mathrm{u}} = \boldsymbol{\sigma}_t^{\mathrm{u},-} \oslash \left( \boldsymbol{\sigma}_t^{\mathrm{u},-} + \boldsymbol{\sigma}_t^{\mathrm{obs}} \right)$$
$$\boldsymbol{q}_t^{\mathrm{l}} = \boldsymbol{\sigma}_t^{\mathrm{s},-} \oslash \left( \boldsymbol{\sigma}_t^{\mathrm{u},-} + \boldsymbol{\sigma}_t^{\mathrm{obs}} \right),$$

where $\oslash$ denotes an elementwise vector division. The update equation for the mean therefore simplifies to

$$\boldsymbol{z}_t^+ = \boldsymbol{z}_t^- + \begin{bmatrix} \boldsymbol{q}_t^{\mathrm{u}} \\ \boldsymbol{q}_t^{\mathrm{l}} \end{bmatrix} \odot \begin{bmatrix} \boldsymbol{w}_t - \boldsymbol{z}_t^{\mathrm{u},-} \\ \boldsymbol{w}_t - \boldsymbol{z}_t^{\mathrm{u},-} \end{bmatrix},$$

where $\odot$ denotes the elementwise vector product. The update equations for the individual parts of covariance are given by

$$\boldsymbol{\sigma}_t^{\mathrm{u},+} = \left( \boldsymbol{1}_m - \boldsymbol{q}_t^{\mathrm{u}} \right) \odot \boldsymbol{\sigma}_t^{\mathrm{u},-},$$
$$\boldsymbol{\sigma}_t^{\mathrm{s},+} = \left( \boldsymbol{1}_m - \boldsymbol{q}_t^{\mathrm{u}} \right) \odot \boldsymbol{\sigma}_t^{\mathrm{s},-},$$
$$\boldsymbol{\sigma}_t^{\mathrm{l},+} = \boldsymbol{\sigma}_t^{\mathrm{l},-} - \boldsymbol{q}_t^{\mathrm{l}} \odot \boldsymbol{\sigma}_t^{\mathrm{s},-},$$

where $\boldsymbol{1}_m$ denotes the $m$ dimensional vector consisting of ones.

## G ROBOTS AND DATA

The experiments are performed on data from two different real robots. The details of robots, data and data preprocessing is explained below:

### G.1 MUSCULOSKELETAL ROBOT ARM

**Observation and Data Set:** For this soft robot we have 4 dimensional observation inputs(joint angles) and 8 dimensional action inputs(pressures). We collected the data of a four DoF robot

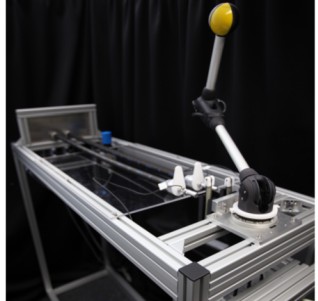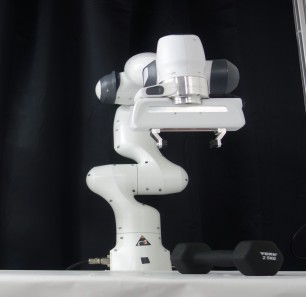

**Figure 7:** The experiments are performed on data from robots with different actuator dynamics. From left to right these include: Pneumatically actuated artificial muscles (Büchler et al., 2016), electrically actuated Franka Emika Panda Robotic Arm.

actuated by Pneumatic Artificial Muscles (PAMs). The robot arm has eight PAMs in total with each DoF actuated by an antagonistic pair. The robot arm reaches high joint angle accelerations of up to 28, 000deg/s2 while avoiding dangerous joint limits thanks to the antagonistic actuation and limits on the air pressure ranges. The data consists of trajectories collected while training with a model-free reinforcement learning algorithm to hit balls while playing table tennis. We sampled the data at 100Hz. The hysteresis associated with the pneumatic actuators used in this robot is challenging to model and is relevant to the soft robotics in general.

**Training Procedure**: For the fully observable case, we trained HiP-RSSM for one-step ahead prediction using an RMSE loss. For partially observable case, during training, we randomly removed half of the observations from the sequences and tasked the models with imputing those missing states, only based on the knowledge of available actions/control commands, i.e., we train the models to perform action conditional future predictions to impute missing states. The imputation employs the model for multi-step ahead predictions in a convenient way.
Other recurrent baselines (RKN, LSTM, GRU) are trained in a similar fashion except that, we don't maintain a context set of interaction histories during training/inference.

### G.2 FRANKA EMIKA PANDA ROBOT ARM

**Observation and Data Set:** We collected the data from a 7 DoF Franka Emika Panda manipulator during free motion and while manipulating loads with weights 0kg (free motion), 0.5 kg, 1 kg, 1.5 kg, 2 kg and 2.5 kg. Data is sampled at high frequencies (1kHz). The training trajectories were motions with loads 0kg(free motion), 1kg, 1.5kg, 2.5 kgs, while the testing trajectories contained motions with loads of 0.5kg and 2 kgs. The observations for the forward model consist of the seven joint angles in radians, and the corresponding actions were joint Torques in Nm. We divide the data into sequences of length 600 while training the recurrent models for forward dynamics, with 300 time-steps (corresponding to 300 milli-seconds) used as context set and rest 300 is used for the recurrent time-series modelling.

**Training Procedure**: For the fully observable case, we trained HiP-RSSM for one-step ahead prediction using an RMSE loss. Similar to the training procedure for partially observable case as in G.1, during training we randomly removed half of the observations(joint angles) from the sequences and tasked the models with imputing those missing observations, only based on the knowledge of available actions/control commands.
Other recurrent baselines (RKN, LSTM, GRU) are trained in a similar fashion except that, we dont maintain a context set of interaction histories during training/inference.

### G.3 WHEELED MOBILE ROBOT

**Observation and Data Set:** We collected 50 random trajectories from a Pybullet simulator a wheeled mobile robot traversing terrain with sinusoidal slopes. Data is sampled at high frequencies (500Hz). 40 out of the 50 trajectories were used for training and the rest 10 for testing. The observations consists of parameters which completely describe its location and orientation of the

robot. The observation of the robot at any time instance $t$ consists of the following features:.

$$o_t = [x, y, z, \cos(\alpha), \sin(\alpha), \cos(\beta)$$
$$\sin(\beta), \cos(\gamma), \sin(\gamma)]$$

where, $x, y, z$ - denote the global position of the Center of Mass of he robot, $\alpha, \beta, \gamma-$ Roll, pitch and yaw angles of the robot respectively, in the global frame of reference (Sonker & Dutta, 2020). We divide the data into sequences of length 300 while training the recurrent models for forward dynamics, with 150 time-steps (corresponding to 300 milli-seconds) used as context set and rest 150 is used for the recurrent time-series modelling.

**Training Procedure**: For the fully observable case, we trained HiP-RSSM for one-step ahead prediction using an RMSE loss. Similar to the training procedure for partially observable case as in G.1, during training we randomly removed half of the observations from the sequences and tasked the models with imputing those missing observations, only based on the knowledge of available actions/control commands.

Other recurrent baselines (RKN, LSTM, GRU) are trained in a similar fashion except that, we dont maintain a context set of interaction histories during training/inference.

## H HYPERPARAMETERS

### H.1 PNEUMATIC MUSCULOSKELTAL ROBOT ARM

**Recurrent Models**

| Hyperparameter | HiP-RSSM | RKN | LSTM | GRU |
|---|---|---|---|---|
| Learning Rate | 8e-4 | 8e-4 | 1e-3 | 1e-3 |
| Latent Observation Dimension | 15 | 15 | 15 | 15 |
| Latent State Dimension | 30 | 30 | 75 | 75 |
| Latent Task Dimension | 30 | - | - | - |

Context Encoder (HiP-RSSM): 1 fully connected + linear output (elu + 1)

- Fully Connected 1: 240, ReLU

Observation Encoder (HiP-RSSM,RKN,LSTM,GRU): 1 fully connected + linear output (elu + 1)

- Fully Connected 1: 120, ReLU

Observation Decoder (HiP-RSSM,RKN,LSTM): 1 fully connected + linear output:

- Fully Connected 1: 120, ReLU

Transition Model (HiP-RSSM,RKN): number of basis: 15

- $\alpha(z_t)$: No hidden layers - softmax output

Control Model (HiP-RSSM,RKN): 3 fully connected + linear output

- Fully Connected 1: 120, ReLU
- Fully Connected 2: 120, ReLU
- Fully Connected 3: 120, ReLU

**Neural Process (NP) Baseline**
Latent Task Dimension: 30
Learning Rate: 9e-4
Optimizer Used: Adam Optimizer

Context Encoder : 1 fully connected + linear output (elu + 1)

- Fully Connected 1: 240, ReLU
- Linear output: 30

Decoder 3 fully connected + linear output

- Fully Connected 1: 512, ReLU
- Fully Connected 2: 240, ReLU
- Fully Connected 2: 120, ReLU

**FFNN Baseline**
Learning Rate: 1e-3
Optimizer Used: Adam Optimizer

2 fully connected + linear output

- Fully Connected 1: 6000, ReLU
- Fully Connected 2: 3000, ReLU

**MAML Baseline**
Meta Optimizer Learning Rate: 3e-3
Inner Optimizer Learning Rate: 0.4
Number Of Gradient Steps: 1
Optimizer Used: Adam Optimizer (Meta Update) and SGD Optimizer (Inner Update)
Model
Feed Forward Neural Network with 3 fully connected + linear output

- Fully Connected 1: 512, ReLU
- Fully Connected 2: 240, ReLU
- Fully Connected 2: 120, ReLU

## H.2 FRANKA ROBOT ARM WITH VARYING LOADS

**Recurrent Models**

| Hyperparameter | HiP-RSSM | RKN | LSTM | GRU |
|---|---|---|---|---|
| Learning Rate | 1e-3 | 1e-3 | 3e-3 | 3e-3 |
| Latent Observation Dimension | 15 | 15 | 15 | 15 |
| Latent State Dimension | 30 | 30 | 75 | 75 |
| Latent Task Dimension | 30 | - | - | - |

Encoder (HiP-RSSM,RKN,LSTM,GRU): 1 fully connected + linear output (elu + 1)

- Fully Connected 1: 30, ReLU

Observation Decoder (HiP-RSSM,RKN,LSTM,GRU): 1 fully connected + linear output:

- Fully Connected 1: 30, ReLU

Transition Model (HiP-RSSM,RKN): number of basis: 32

- $\alpha(z_t)$: No hidden layers - softmax output

Control Model (HiP-RSSM, RKN): 1 fully connected + linear output

- Fully Connected 1: 120, ReLU

**Neural Process (NP) Baseline**
Latent Task Dimension: 30
Learning Rate: 1e-3
Optimizer Used: Adam Optimizer

Context Encoder : 1 fully connected + linear output (elu + 1)

- Fully Connected 1: 240, ReLU
- Linear output: 30

Decoder 3 fully connected + linear output

- Fully Connected 1: 512, ReLU
- Fully Connected 2: 240, ReLU
- Fully Connected 2: 120, ReLU

**FFNN Baseline**
Learning Rate: 1e-3
Optimizer Used: Adam Optimizer

2 fully connected + linear output

- Fully Connected 1: 6000, ReLU
- Fully Connected 2: 3000, ReLU

**MAML Baseline**
Meta Optimizer Learning Rate: 3e-4
Inner Optimizer Learning Rate: 0.04
Number Of Gradient Steps: 1
Optimizer Used: Adam Optimizer (Meta Update) and SGD Optimizer (Inner Update)
Model
Feed Forward Neural Network with 3 fully connected + linear output

- Fully Connected 1: 512, ReLU
- Fully Connected 2: 240, ReLU
- Fully Connected 2: 120, ReLU

H.3 WHEELED ROBOT TRAVERSING SLOPES OF DIFFERENT HEIGHT

**Recurrent Models**

| Hyperparameter | HiP-RSSM | RKN | LSTM | GRU |
|---|---|---|---|---|
| Learning Rate | 9e-4 | 9e-4 | 1e-2 | 1e-2 |
| Latent Observation Dimension | 30 | 30 | 15 | 15 |
| Latent State Dimension | 60 | 60 | 75 | 75 |
| Latent Task Dimension | 60 | - | - | - |

Encoder (HiP-RSSM,RKN,LSTM,GRU): 1 fully connected + linear output (elu + 1)

- Fully Connected 1: 120, ReLU

Observation Decoder (HiP-RSSM,RKN,LSTM,GRU): 1 fully connected + linear output:

- Fully Connected 1: 240, ReLU

Transition Model (HiP-RSSM,RKN): number of basis: 15

- $\alpha(\boldsymbol{z}_t)$: No hidden layers - softmax output

Control Model (HiP-RSSM, RKN): 3 fully connected + linear output

- Fully Connected 1: 120, ReLU

**Neural Process (NP) Baseline**
Latent Task Dimension: 30
Learning Rate: 1e-3
Optimizer Used: Adam Optimizer

Context Encoder : 1 fully connected + linear output (elu + 1)

- Fully Connected 1: 240, ReLU
- Linear output: 30

Decoder 3 fully connected + linear output

- Fully Connected 1: 512, ReLU
- Fully Connected 2: 240, ReLU
- Fully Connected 2: 120, ReLU
- Fully Connected 2: 60, ReLU

**FFNN Baseline**
Learning Rate: 2e-3
Optimizer Used: Adam Optimizer

3 fully connected + linear output

- Fully Connected 1: 512, ReLU
- Fully Connected 2: 240, ReLU
- Fully Connected 2: 120, ReLU
- Fully Connected 2: 60, ReLU

**MAML Baseline**
Meta Optimizer Learning Rate: 3e-4
Inner Optimizer Learning Rate: 0.4
Number Of Gradient Steps: 1
Optimizer Used: Adam Optimizer (Meta Update) and SGD Optimizer (Inner Update).
Model
Feed Forward Neural Network with 3 fully connected + linear output

- Fully Connected 1: 512, ReLU
- Fully Connected 2: 240, ReLU
- Fully Connected 2: 120, ReLU
- Fully Connected 3: 60, ReLU

