# OpenReview forum: "Hidden Parameter Recurrent State Space Models For Changing Dynamics Scenarios"
_ICLR.cc/2022/Conference — ICLR 2022 Poster_

### Official Review · Reviewer_nyLc · 2021-10-27

**Correctness:** 3
**Technical Novelty And Significance:** 3
**Empirical Novelty And Significance:** 3
**Recommendation:** 6
**Confidence:** 3

**Main Review:**


#### Pros
- Moderate extension of existing recurrent state space models for changing dynamics for well-motivated problems.
- Detailed experimental configurations are in the appendix.
- Applications on different robot learning problems and settings are evaluated.

#### Cons
- For the experiments with partial observations and missing values, the authors may need to compare with some RNN baselines designed for handling partial observations/missing values.
- The definition and scope of the context set $C_l$ and some related contents are quite ambiguous to the readers, which disrupts the understanding and evaluation of the proposed method.
  - Based on the definition in Section 3.2.1, $C_l$s are different for different time step $t$; Meanwhile, $l$ is fixed for the duration of the task.
  - In Section 3.2.1: $T$ has $N$ time steps. While in Section 3.3 $T$ is of length $M$. Please explain the differences between these variables.
  - $t \in T \in R^{B \times M \times D}$. What is $t$?

#### Details
- In the second paragraph of the introduction, the authors motivate this work by listing both time-invariant ("similar but not identical") dynamics and time-variant dynamics (such as the friction changes and wear and tear). How are these two different types of dynamics captured in the proposed model, in the same way or with varying model formations? The experiments conducted in Section 5.1 are assumed to be for the time-variant dynamics, but more evidence about this assumption is needed.
- However, the proposed model seems only to handle the former ones. Also, the experiments are not conducted with the time-changing dynamical systems.
- It is unclear what the concept of latent task refers to (could the authors provide some examples?) and why multi-task models are suitable to compare as baselines.

- Some definitions and concepts in Section 2.2 are not clear until the readers reach Section 3.2. Content rearrangements or clarifications are needed.
- Similarly, it would be suggested to elaborate more model details about, e.g., locally linear transition.

- Minur issues:
  - Introduction: "... like images was ..." -> "... like images and was ..."
  - Section 3.2: "Each of these are three stages ..." -> "Each of these three stages ..."
  - Section 3.2.3: "this a Gaussian conditioning layer" -> "this is a ..."

**Summary Of The Paper:**

The authors extend the recurrent Kalman networks with hidden parameters and context set features to model changing dynamics. The authors demonstrate the performance of the proposed method with a series of experiments on robot learning.

**Summary Of The Review:**

This paper provides plausible and moderate extensions to the existing model to solve problems in a new setting. However, some necessary clarifications and important baselines are missing. Therefore, I'd like to put it borderline and am willing to adjust my scores if more information from the rebuttal and discussion phases is provided.

---

> ### Author Response · Authors · 2021-11-21
> **Author Reply (Part 1)**
>
> **Concern 1:** *“For the experiments with partial observations and missing values, the authors may need to compare with some RNN baselines designed for handling partial observations/missing values.”*
>
> **Reply** : We in fact compare RNN baselines that are designed for handling missing values. We compare our method with a deep SSM (/http://proceedings.mlr.press/v97/becker19a/becker19a.pdf) which was designed to deal with missing observations where they skip the posterior update and use the prior when encountering missing observations. Similarly our deterministic RNN baselines LSTM/GRUs have an encoder decoder architecture (more details in appendix) to deal with partial observability/missing values. The architecture for LSTMs/GRUs is based on similar baselines used for missing value/parital observability experiments in http://proceedings.mlr.press/v97/becker19a/becker19a.pdf.
>
> **Concern 2:**  *“The definition and scope of the context set $C_l$
>  and some related contents are quite ambiguous to the readers, which disrupts the understanding and evaluation of the proposed method……..”*
>
> **Reply:** We apologize for the missing details regarding context set construction which may confuse the readers. We have addressed this concern to the best of our ability in the updated draft. We detail this in Section 3.3, how HiP-RSSM is trained for multi-task learning with further details on context set creation explained in Appendix B and C. $C_l$ is fixed for the entire duration of the task(i.e. a short time segment of the trajectory). Further clarifications were made in sections 3.2.1 and 3.3. $t \in T$ in section 3.3 refers to a local temporal segment over which we assume the local consistency in dynamics.
>
> M in section 3.3 refers to the length of this target time series (local temporal segment), while N refers to the length of the context set. While in principle the context and target sets can be of different lengths, we have chosen the length of context and target sets to be the same in all our experiments (also mentioned in Appendix H). To avoid confusion we have replaced M with N in section 3.3.
>
> **Concern 3:** *“It is unclear what the concept of latent task refers to (could the authors provide some examples?) and why multi-task models are suitable to compare as baselines”.*
>
> **Reply:** Multi-task settings for changing dynamics scenarios with the definition of “task” as a local temporal segment have been explored previously with the gradient-based meta learning algorithm of MAML (/https://arxiv.org/pdf/1803.11347.pdf). This has also been discussed in the related works section under “Meta-Learning For Changing Dynamics”. Thus we also compare our algorithms with multi-task baselines.
>
> We detail this multi-task learning formalism at multiple sections in the updated draft. Firstly we detail in Section 3.3, how HiP-RSSM is trained for multi-task learning where each task is defined as a local dynamics in a temporal segment of the trajectory. The distribution over these tasks/local dynamical systems is modelled using a hierarchical latent task variable $l$. Secondly, we show how a multi-task model trained to capture varying dynamics can perform inference over a long trajectory with changing dynamics in Appendix B. Our formalism can capture the global changes in dynamics since we obtain a different instance of the HiP-RSSM for each temporal segment based on the observed context set.
>
> **Concern 4:**  *“In the second paragraph of the introduction, the authors motivate this work by listing both time-invariant ("similar but not identical") dynamics and time-variant dynamics (such as the friction changes and wear and tear). How are these two different types of dynamics captured in the …… model seems only to handle the former ones. Also, the experiments are not conducted with the time-changing dynamical systems.”*
>
> **Reply:** We capture this with the same multi-task formalism which captures the variation in the dynamics of different instances (local temporal segments) through a set of hidden task parameters. For capturing globally changing dynamics we follow an inference procedure mentioned in Appendix B in the updated draft, where we obtain a different instance of the HiP-RSSM for each temporal segment based on the observed context set. We also add an additional experimental section (section 5.4) where we plot the global changes in hidden parameters (see Figure 4b and 4c) captured using HiP-RSSM, when performing inference over a long trajectory of varying dynamics.

---

> > ### Author Response · Authors · 2021-11-21
> > **Author Reply (Part 2)**
> >
> > **Concern 5:** *”Some definitions and concepts in Section 2.2 are not clear until the readers reach Section 3.2. Content rearrangements or clarifications are needed.”*
> >
> > **Reply:** We have more clarifications in section 3.2. However, we decide to keep section 2.2 unchanged since it discusses a previously proposed method. More changes can be made if the reviewer further insists.
> >
> > **Concern 6:** *”Similarly, it would be suggested to elaborate more model details about, e.g., locally linear transition.”*
> >
> > **Reply:** More intuitions about the locally linear transition have been added in Appendix F3. Architecture details for $A_t$ including number of basis matrices used can be found in Appendix H.
> >
> > The minor issues raised were also taken care of. Thanks for pointing these out.
> >
> > We hope we could clarify/remove most of your concerns. We invite you to ask additional questions and engage in further discussion if this is not the case.

---

> > > ### Author Response · Authors · 2021-11-29
> > > **Request For Feedback**
> > >
> > > Dear Reviewer,
> > >
> > > Many thanks for your valuable comments and reviews. We hope that you've had a chance to read our response and the updated version of the paper. We would really appreciate a reply as to whether our response and clarifications have addressed the issues raised in your review, or whether there is anything else we can address.
> > >
> > > Authors

---

> > > > ### Comment · Reviewer_nyLc · 2021-11-30
> > > > **Thanks for the reply**
> > > >
> > > > The reviewer has read the other reviews and the responses and would like to thank the authors for the informative replies.
> > > > A majority of the questions have been clarified, while the absence of advanced imputation baseline comparison and some unclear descriptions in Sec. 2.2 remain.
> > > > Meanwhile, the author proposed the new concept of the ``local temporal segment of the trajectory'' in the revision and rebuttal, which is not mentioned in the original paper. This concept sort of limits the method's capacity from learning general changing dynamics to learning local dynamics.
> > > > Hence, the reviewer tends to keep the original score.

---

> > > > > ### Author Response · Authors · 2021-12-01
> > > > > **Thank You For Your Feedback**
> > > > >
> > > > > Dear Reviewer,
> > > > >
> > > > > We thank you for going over our response and the updated draft. We appreciate your positive score and response.
> > > > >
> > > > > We would like to clarify a misunderstanding that a new concept was introduced in the rebuttal draft that was different from the original draft. We in fact make the same assumption of a HiP-MDP formalism (/https://www.ijcai.org/Proceedings/16/Papers/206.pdf) which forms the basis for our HiP-SSM formalism introduced in section 3.1, that each new episode(local trajectory) can be considered as a new instance of an MDP(in HiP-MDP formalism) or an SSM(in our HiP-SSM formalism). So each local trajectory/episode is a new task/SSM/MDP. And we learn a distribution over these tasks with the latent variable $l$ using our training procedure (which was written more clearly in the updated draft). Similar assumptions are also made in the reference paper(/https://arxiv.org/pdf/2006.10701.pdf) reviewer VmRd suggested, which also assumes local consistency in dynamics over local episodes and yet shows that it can handle non-stationary environments. We formalize this setting for recurrent state-space models.
> > > > >
> > > > > The local consistency of dynamics assumption and our HiP-SSM formalism is in no way limiting our model’s capacity to handle non-stationary dynamics that an agent might encounter in its lifetime. It can handle time-varying dynamics by inferring a new $l$ for each local trajectory/episode based on the observed context set. In most robotic settings one can’t expect the dynamics to change rapidly over these short local trajectories/episodes (less than 1 second in all our experiments). Our newly added experiments show how HiP-RSSM formalism can handle both discrete (Franka manipulator) and continuous (mobile robot traversing a terrain of different slopes) changes in dynamics globally over the long trajectories(many seconds long) that an agent encounters in its lifetime in our updated draft (section 5.4 and figure 4b and 4c).
> > > > >
> > > > > We hope this misunderstanding is clarified. We apologize if this was not clear in the initial draft. Also, we are willing to rewrite section 2.2 clearly and add new imputation baselines in a final draft. Thanks again for your reviews and valuable suggestions.

---

> > > > > > ### Comment · Reviewer_nyLc · 2021-12-02
> > > > > > **Thanks you for the follow-up response**
> > > > > >
> > > > > > Thank you for your quick follow-up response for clarifying the local temporal segment and dynamics.
> > > > > > I encourage and expect the updates/revisions mentioned by the authors in the final draft or any further version of the paper.

---

### Official Review · Reviewer_8V7G · 2021-11-02

**Correctness:** 2
**Technical Novelty And Significance:** 1
**Empirical Novelty And Significance:** 2
**Recommendation:** 5
**Confidence:** 4

**Main Review:**

### Strengths
Extending current state-space models to handle changing dynamics is important and the problem being addressed in this work is relevant to the community. The paper describes the approach reasonably well (up to the points below). A plus point is that the authors conducted experiments using real robots to demonstrate their approach can be applied to these settings.

### Weaknesses
At present, technical novelty is unclear in that HiP-RSSM appears to be a variation of existing models. The primary equations are derived from the standard Kalman filter and there doesn't appear to be new insights or technical advances in the method's derivation. The experiments are not clearly described in the main text, which makes it difficult to validate the main claim that HiP-RSSM works better than existing approaches. Please see below for details.

- The proposed model bears resemblance to the Kalman VAE (KVAE) [Fraccaro et al. 2017] which is not cited but highly related. The key difference is the introduction of the latent variable $l$. Initially, I expected that the transition matrices $A_t$ or control function to be dependent on $l$ but the variable enters only as an additive term in the updates. As an aside, I suggest that B.5 be brought forward into the main text since it is a core part of the method.
- The proposed model is also rather confusing since the mean $\mu_l$ is used in these updates rather than $r^l_n$? Can the authors better motivate the need to introduce the latent variable in this manner? How would it compare to a simple baseline that modifies the latent representation with an extra bias dimension and learns a time-varying transition matrix $A_t$?
- The literature review also misses out on related work on switching dynamical systems, e.g., state-dependent or explicit duration models, which can capture the change in dynamics. This work appears to be a special case whereby the switching is not explicitly captured and technically contribution appears limited.
- Is the training in the experiments done using RMSE or using eqn (9)? If I understand the paper, it seems like the method is training solely using RMSE which makes the proposed probabilistic setup appear unnecessary.
- The prediction experiments results are for a one time-step prediction (only explained in the supplementary) and it's unclear how meaningful the results are. Typically, we are interested in predictions over long horizons, e.g., for MPC. This also motivates the need for capturing the change in dynamics (e.g., in the switching models) that is modelled by the HiP-RSSM.

References:

Fraccaro, Marco, et al. "A disentangled recognition and nonlinear dynamics model for unsupervised learning." NeurIPS (2017).

**Summary Of The Paper:**

This work proposes to extend latent state-space models (SSMs) with a latent variable that changes the dynamics. Update equations akin to Kalman filtering are provided, along with a training loss and method. Experiments on several robotics tasks appear to indicate that the method performs well relative to alternative methods that do not consider latent dynamics differences (up to the points below).

**Summary Of The Review:**

While I do believe that the paper has some merits, the technical contribution appears limited and I am unable to recommend an accept. It would be helpful if the authors clarify differences to existing models, especially KVAE and switching state-space models.

---

> ### Author Response · Authors · 2021-11-21
> **Author Reply (Part 1)**
>
> **Concern 1** *”The proposed model bears resemblance to the Kalman VAE (KVAE) [Fraccaro et al. 2017] which is not cited but highly related. The key difference is the introduction of the latent variable . Initially, I expected that the transition matrices
>  or control function to be dependent on  but the variable enters only as an additive term in the updates. As an aside, I suggest that B.5 be brought forward into the main text since it is a core part of the method.”*
>
> **Reply:** Thank you for pointing out this missing related work… We have cited KVAE in our related works section and also at places where locally linear models have been used… However, we use a more recent Deep-SSM (/http://proceedings.mlr.press/v97/becker19a/becker19a.pdf) to baseline our model as becker et al. has shown better empirical performance compared to KVAE in their experiment section… Moreover, KVAE uses computationally expensive matrix inversions and hence we believe would not scale to the real robot settings with high-frequency data as used in our experiment… We have added parts of F.5 (in updated drat) into the main text in section 3.2.2 as suggested by the reviewer.
>
>
> **Concern 2** *”The proposed model is also rather confusing since the mean $\mu_l$
>  is used in these updates rather than $r_n^l$. ? Can the authors better motivate the need to introduce the latent variable in this manner?”*
>
> **Reply:** $r_n^l$ is a single element of the latent context set consisting of interaction histories over previous N time steps… while $\mu_l$ is a single vector that, that intuitively acts as a ‘summary’ of this context set elements obtained by weighted averaging based on bayesian principles… So it does not make sense to use $r_n^l$ instead of $\mu_l$… These latent task representations based on deepset functions has been inspired from recent works in Neural Processes and can be interpreted as learning ‘deep kernels’ in our scheme (/http://bayesiandeeplearning.org/2018/papers/128.pdf ) that learns a distribution over tasks... Moreover, we add an additional experiment that signifies the importance of this permutation invariant set-based representation in Appendix E.
>
>
> **Concern 3** *“The literature review also misses out on related work on switching dynamical systems, e.g., state-dependent or explicit duration models, which can capture the change in dynamics. This work appears to be a special case whereby the switching is not explicitly captured and technically contribution appears limited.”*
>
> **Reply:** We have added a section for ‘recurrent switching dynamical systems’ in section 4 of the updated draft, where we highlight the differences. We take a rather different formalism compared to these switching models which use discrete hierarchical latent variables to capture multi-modal/changing dynamics. We view the problem as a multi-task learning task with a continuous hierarchical variable that models the distribution over these tasks. With our formalism which considerably simplifies the procedure for inferring the hidden parametrization, we are still able to capture the global evolution of parameters rather efficiently as seen in figures 4b and 4c in the updated draft.  However, learning dynamics over these hidden task parameters can be taken up as future work and may lead to better generalization and inter-task knowledge transfer.
>
> **Concern 4** *“At present, technical novelty is unclear in that HiP-RSSM appears to be a variation of existing models. The primary equations are derived from the standard Kalman filter and there doesn't appear to be new insights or technical advances in the method's derivation.”*
>
> **Reply:** The formalism extending deep SSMs to multi-task learning is novel and doesn’t exist in prior literature to the best of our knowledge. The derivations for task conditional kalman time update in section 3.2.2 are novel and do not exist in prior works. Moreover, we highlight in the related works section how our method differs from current literature on switching dynamical systems in the updated draft. Our formalism is simpler, principled and is scalable enough to be deployed on real robotic tasks under changing dynamics scenarios.
>
> **Concern 5** *”Is the training in the experiments done using RMSE or using eqn (9)? If I understand the paper, it seems like the method is training solely using RMSE which makes the proposed probabilistic setup appear unnecessary.”*
>
> **Reply:** Yes, we use an RMSE loss. We have stated this explicitly in the updated draft and the probabilistic formulation has been moved to the appendix. We could train in principle with a likelihood loss however training with RMSE yields slightly better performance in our experiments and allows a fair comparison with deterministic baselines like Feed-forward NN, LSTM, GRUs and MAML and we report results with RMSE.

---

> > ### Author Response · Authors · 2021-11-21
> > **Author Reply (Part 2)**
> >
> > **Concern 6** *“How would it compare to a simple baseline that modifies the latent representation with an extra bias dimension and learns a time-varying transition matrix
> > ?”*
> >
> > **Reply** Our architecture is based on closed form updates that we obtain via forward inference algorithm. It is a principled way to achieve latent task conditioning. Nevertheless our preliminary experiments with a latent task mean ($\mu_l$) conditioned locally linear transition model whose coefficient network $\alpha$ now gets the $\mu_l$ as an input, gave significantly worse performance especially for multi step ahead predictions. The empirical results can be added to the appendix if the reviewer insists.
> >
> > **Concern 7** *”The prediction experiments results are for a one time-step prediction (only explained in the supplementary) and it's unclear how meaningful the results are. Typically, we are interested in predictions over long horizons, e.g., for MPC. This also motivates the need for capturing the change in dynamics (e.g., in the switching models) that is modelled by the HiP-RSSM.”*
> >
> > **Reply** We add an additional section in the Appendix E where we show how the model fares at multi step ahead predictions.
> >
> > We hope we could clarify/remove your concerns. We invite you to ask additional questions and engage in further discussion if this is not the case.

---

> > > ### Author Response · Authors · 2021-11-29
> > > **Request For Feedback**
> > >
> > > Dear Reviewer,
> > >
> > > Thanks again for your valuable comments and reviews. We hope that you've had a chance to read our response and the updated version of the paper. We would really appreciate a reply as to whether our response and clarifications have addressed the issues raised in your review, or whether there is anything else we can address.
> > >
> > > Authors

---

> > > > ### Comment · Reviewer_8V7G · 2021-11-29
> > > > **Thanks for the response!**
> > > >
> > > > Thank you for the response and the additional content. I have read it and the other reviews. I remain of the opinion that the problem addressed is interesting but the proposed model has limited novelty given existing work in deep Kalman models and switching dynamical systems. Also, training via RMSE also seems to be at odds with the PGM setup and relevant derivations, making the formulation less principled.

---

> > > > > ### Author Response · Authors · 2021-12-01
> > > > > **Thank You For Your Feedback**
> > > > >
> > > > > Dear Reviewer,
> > > > >
> > > > > Thank you for your review and feedback. We hope the reviewer will look at this work in the context of extending SSMs (including deep Kalman Models) to multi-task learning / non-stationary dynamics scenarios.
> > > > >
> > > > > Authors

---

### Official Review · Reviewer_vXB5 · 2021-11-02

**Correctness:** 3
**Technical Novelty And Significance:** 2
**Empirical Novelty And Significance:** 3
**Recommendation:** 8
**Confidence:** 4

**Main Review:**

# Strengths

I am a huge fan of this work as I think a lot of nifty tricks are used. Firstly, I thought the introduction and the motivation of this work is top-notch! Secondly, the form of the generative model is very elegant as it allows for a global dynamics model to be learned that can be shared across tasks/environments while the context variable allows for specialization within a task (and is also very easy to compute at test-time).
The use of "linear" (I will have comments on this later) dynamics allows for everything to computed in closed-form as well. To avoid issues with likelihoods that are parameterized by NNs, the authors also use a pseudo-like likelihood where a separate encoder is used to project the observations to some feature space that is used to inform the latent states and while retaining the closed-form updates of Kalman filtering.

# Weaknesses

I think there are very important details that are swept under the rug.
Firstly, the authors use a locally-linear transition model
$$ A_{t-1} = \sum_{k=1}^K \alpha(z_{t-1}) A^{k} $$
where $\alpha(\cdot)$ is a softmax parameterized by a NN.
Note that while Kalman filtering can handle time-varying linear dynamics, they have to be *indepedent* of $z_{t-1}$.
This parameterization *prevents* the use of Kalman filtering as now $z_t$ nonlinearly depends on $z_{t-1}$! This is a well-known problem that has been attempted to be addressed in the switching linear dynamical systems literature i.e.,  (https://proceedings.mlr.press/v54/linderman17a.html, https://arxiv.org/abs/1811.12386).

Secondly, the marginalization of the context variable, $l$, is only available in closed form if the $z_t$ depends linearly on $l$. But in some of the experiments, the authors pass the context variable through a NN, preventing closed-form marginalization; they instead take an approximation, passing in the mean and variance of the through the NN instead (I'm somewhat confused by the passing the variance through the NN. By the principle of propagation of uncertainty, it should be something like the derivative squared times the variance, https://en.wikipedia.org/wiki/Propagation_of_uncertainty).
While I'm somewhat fine with these two approximations as it allows for ease of computations, the authors should explicitly state this and avoid the use of "exact inference". Moreover, I would love to see some sort of empirical evidence that taking these approximations is okay i.e. comparing their KF to an extended KF.

Lastly, one other detail that is missing is the discussion of the context sets. The description is somewhat confusing. On page 3, the authors state "We also make the additional assumption that the parameter vector $l$ is fixed for the duration of the task, and thus the latent task parameter has no dynamics".
But on page 4, the parameterization and description of the context set makes it seem that one is inferring a sequence of $l$'s,
"for any time series $\mathcal{T} = (o_t, a_t, o_{t+1}, ..., o_{t+N}, a_{t+N})$, the context set $C_l$ consists of tuples of
the current state/observation, current action and next state/observation for the previous N time steps, i.e. $C_l = (o^l_{t-n}, a_{t-n}^l, o_{t-n+1}^l)_{n=1}^N $".
What I am assuming is happening is that the data is split a priori into a context set and another set used for performing the Kalman filtering updates.
I suggest using a separate subscript for the context sets.
Also, the constructions of the context sets don't seem to be discussed: what is $N$, how were the context sets collected, etc.

# Comments/Questions
1. On equation &, it should be $z_t^- = A_{t-1} z_{t-1} + Ba_t + C\mu_l$
2. Many models can perform decently well on 1-step ahead prediction (for instance, an LDS can perform well at one-step ahead prediction if the dynamics are slow, see Figures 2 and 3 in https://arxiv.org/abs/1811.12386). Have you investigated how your model fares on multi-step ahead prediction?


------------------------
# Update
I think the authors have done a good job responding to my concerns and the concerns of the other reviewers. As such, I have raised my score to an 8.

**Summary Of The Paper:**

This paper introduces a state-space model for non-stationary environments (where in this formulation the latent dynamics are changing while the observation model is fixed), which they call Hidden Parameter Recurrent State Space Models (HiP-RSSMs). In HiP-RSSMs, a context variable is inferred which is used to modify the latent dynamics to adapt towards the new setting.

**Summary Of The Review:**

I think the idea presented in the paper is great and well-motivated. The approach is elegant and the empirical results are compelling. Most of my concerns are in--important--missing details and details that are swept in the appendix, which I think can be easily fixed with some rewriting. For this reason, I will give a marginal acceptance. I am more than happy to increase my score if the authors address my concerns.
Also, if I misunderstood/missed something please let me know!

---

> ### Author Response · Authors · 2021-11-21
> **Author Reply**
>
>
> Thank you for your review and comments. We will try to address some of the concerns you raised here.
>
> **Concern 1**: *Firstly, the authors use a locally-linear transition model…………...they have to be independent of $z_{t-1}$. This parameterization prevents the use of Kalman filtering as now $z_{t-1}$ nonlinearly depends on $z_t$*
>
> **Reply:** Sorry, there was a typo. Its $z_t^+$ and not $z_t$, where $z_t^+$ is the notation for the posterior mean in the main text. We have made this correction in the updated draft. The locally linear model is defined as follows, $A_t = \sum_{k=0}^K \alpha^{(k)}(z_t^+) A^{(k)}$, where $z_t^+$ is the posterior mean and not a random variable/sample. So we have a linearized $A_t$ around the posterior mean, which can be interpreted as an EKF. Since $A_t$ does not depend explicitly on the random variable $z$, but only the mean, the linear dependency between consecutive states of $z$ is preserved. This parameterization of the locally linear model has been used in previous works (https://proceedings.mlr.press/v97/becker19a.html) .
>
> **Concern 2:** *Secondly, the marginalization of the context variable l, ……………….. While I'm somewhat fine with these two approximations as it allows for ease of computations, the authors should explicitly state this and avoid the use of "exact inference".*
>
> **Reply:** We agree this neural network that matches the moments is an approximation, but works well in practice compared to the more principled linear / locally linear transformations as shown in our ablations (figure 3c). This has also been added in the main text explicitly. We also replaced the term “exact inference” with “inference with forward algorithm”, since these additive interactions and network architecture are inspired from the closed form solutions we get via forward inference algorithm.
>
> **Concern 3**: *Lastly, one other detail that is missing is the discussion of the context sets………………. Also, the constructions of the context sets doesn't seem to be discussed: what is $N$, how were the context sets collected, etc.*
>
> **Reply:** We apologize for the missing details regarding context set construction and this has been addressed to the best of our ability in the updated draft at several places.  We detail this in Section 3.3, how HiP-RSSM is trained for multi-task learning with further details on context set creation explained in Appendix B and C. Further clarifications were made in section 3.2.1. The context / target creation is in line with the reviewer's assumption, where the data is split a priori into a context set and another (target) set used for performing the Kalman filtering updates.
>
>  $N$ is the number of samples over the target temporal segment over which we are performing Kalman Filtering Updates. These temporal segments are chosen to be of the length 0.3 to 0.4 seconds depending upon the robots (Appendix G). These are chosen in line with the horizon length used for planning in model-based RL.
>
>
> **Question 1:**  *”On equation (7), it should be ….”*
>
> **Reply:** Thanks for pointing this out. We have made this correction.
>
> **Question 2:** *”Have you investigated how your model fares on multi-step ahead prediction?”*
>
> **Reply:** We add an additional section in Appendix E where we show how the model fares at multi-step ahead predictions.
>
> We hope we could clarify/remove most of your concerns. We invite you to ask additional questions and engage in further discussion if this is not the case.

---

> > ### Author Response · Authors · 2021-11-29
> > **Request For Feedback**
> >
> > Dear Reviewer,
> >
> > Thanks again for your insightful comments and reviews. We hope that you've had a chance to read our response and the updated version of the paper. We would really appreciate a reply as to whether our response and clarifications have addressed the issues raised in your review, or whether there is anything else we can address.
> >
> > Authors

---

> > > ### Comment · Reviewer_vXB5 · 2021-11-29
> > > **Response**
> > >
> > > Thank you so much for the updates and I vehemently apologize for my late response!
> > >
> > > After reading the author's responses to all of the author's concerns, I will raise my score to an 8 as I think the authors have done a good job responding to most of the concerns brought up.

---

> > > > ### Author Response · Authors · 2021-12-01
> > > > **Thanks For The Feedback**
> > > >
> > > > Dear Reviewer,
> > > >
> > > > Many thanks for your positive feedback and updated score.
> > > >
> > > > Authors

---

### Official Review · Reviewer_VmRd · 2021-11-02

**Correctness:** 2
**Technical Novelty And Significance:** 2
**Empirical Novelty And Significance:** 2
**Recommendation:** 5
**Confidence:** 4

**Main Review:**

The core idea leverages the model specification of a hidden-parameter-MDP (HiP-MDPs)  such that the hidden parameter encodes the time-varying aspect of the dynamics.  If I understand it right, the model considered overall consists of a latent variable (for e.g., motor friction, which is perhaps globally changing but locally fixed), a true underlying state (for e.g., exact location of the robot which changes locally as well), and an observation (for e.g., image of where the robot is).


- Why are permutation invariant deep models used for encoding the history of past N steps? Why is it useful to discard the temporal information?

- Section 3.2.2: It is mentioned that $z_{t-1},l, a$ are independent of each other as $z_t$ is unobserved. How are $z_{t-1}$ and $l$ independent. From Figure 2, does not $l$ directly influences $z_{t-1}$?

- Section 3.2.2: what is the random variable being sampled from the third Gaussian distribution? Is it supposed to be $y$ instead of $v$?

- Section 3.2.3: It might be better to provide some information in the main paper about how inverse on the term that depends on H is being done in a computationally efficient manner.


- The key idea of the proposed work is to provide a wrapper around the Kalman filter. Exploiting the fact that the hidden-parameter is fixed throughout the task, the proposed method used deep nets to infer the hidden-parameter. This inferred hidden parameter is then used along with the standard state-space model, where the next state changes linearly in the past state, action, and the hidden parameter.

- If I understand it right, in the proposed setup, deep nets were possible for inferring the hidden-parameter as the hidden parameters never change and thus do not need any posterior updates. Whereas, the state does evolve and thereby necessitating linear evolution structure.

- Further, because the method builds around Kalman filtering, it is assumed that all the noises are isotropic Gaussian, such that covariances are zero and Kalman updates can be done easily. I am not sure why this assumption is reasonable in general.


- As the paper title says "changing dynamics scenario", perhaps it is only reasonable to assume that the proposed method will handle that. While it is fine to assume that locally the hidden parameter is fixed (implying dynamics is fixed), ultimately it needs to be shown how at the global level the method can track the changing dynamics parameter as well.

  - Currently, both in the model setup and the experiments, this setup is missing. The model assumes that the hidden parameter value is fixed throughout and does not consider at all _how the current hidden parameter influences the next value of the hidden parameter_.

  - If I understand it right, even in the experiments constructed, each observation sequence is generated using a fixed value of the hidden parameter. Notice that this is not a fair comparison with the baseline methods. For example, even the basic GRU and LSTM models are more general as they can consider the latent variable to be a part of state-space that is evolving with time.

- Further, the paper misses out on recent works by Xie et al. (2020) that build upon the work by Nagabandi et al. (2018a;b) and Finn et al. (2017). They look at a very similar HiP-MDP setup (and also goes beyond just modeling the time-varying dynamics and tries to do control). While they make the assumption of dynamics having a locally fixed hidden parameter like this work, they also consider how the hidden parameter is evolving more globally.

Deep Reinforcement Learning amidst Lifelong Non-Stationarity.
Annie Xie, James Harrison, Chelsea Finn.
International Conference on Machine Learning (ICML), 2021.

- Under related works, it is mentioned that existing deep state-space models assume stationarity but their proposed method does not. I do not think this is true. If one considers the latent variable to be sampled from a starting distribution and held fixed throughout the episode (as what the proposed method has done both in experiments and the methodology), then it can be seen that everything is just stationary as well and warrants comparison with the existing deep state-space methods.

-----
Minor

- The preliminary section builds up the notations as if arbitrary non-linear functions will be used later to model the dynamics, which is not completely true.

- I think the paper can benefit a lot by clarifying the relations between $x$, $w$, $o$, and also between $\eta$, $z$, $l$,

Typos  Section 3.2.2:
- Figure 1 => Figure 2?
- Should $z_t^{-}$ depended on $z_{t-1}^-$ instead of $z_{t-1}$?
- Is the matrix A time-varying or fixed? Notations are used inter-changeably.




**Summary Of The Paper:**

The paper proposes a method to learn a probabilistic recurrent state-space model for time-varying dynamics. The proposed method combines the Kalman filtering-based update rule with deep network-based encoder and decoder model. Effectively, the method can be used to replace RNN cells in a recurrent model, and is shown to outperform baseline models in modeling various robotic tasks.

**Summary Of The Review:**

Overall, the proposed method claims to be designed for environments that are dynamically changing over time, but actually assumes the latent variable to be fixed throughout the entire episode (not just locally for a small part of the episode). As such it contradicts the main claim.

Let me know if my understanding of the work is not accurate and I will be happy to reconsider the score.

---

> ### Author Response · Authors · 2021-11-21
> **Author Reply (Part 1)**
>
> We thank the reviewer for the detailed comments and reviews. Please find below the reply to the different concerns that you pointed out.
>
> **Concern 1**: *As the paper title says "changing dynamics scenario", perhaps it is only reasonable to assume that the proposed method will handle that. While it is fine to assume that locally the hidden parameter is fixed (implying dynamics is fixed), ultimately it needs to be shown how at the global level the method can track the changing dynamics parameter as well.*
>
> **Reply**: We agree that the initial draft was not clear on how HiP-RSSM can model an agent’s dynamics that changes continuously over time. This has been addressed in the updated version of the paper at several places. Firstly we detail in Section 3.3, how HiP-RSSM is trained for multi-task learning where each task is defined as a local dynamics in a temporal segment of the trajectory. Secondly, we show how a multi-task model trained to capture varying dynamics can perform inference over a long trajectory with changing dynamics in Appendix B. Lastly, as requested by the reviewer we also add an additional experimental section (section 5.4) where we plot the global changes in hidden parameters (see Figure 4b and 4c) using HiP-RSSM.
>
> **Concern 2**:*“ Overall, the proposed method claims to be designed for environments that are dynamically changing over time, but actually assumes the latent variable to be fixed throughout the entire episode (not just locally for a small part of the episode). As such it contradicts the main claim.”*
>
> **Reply**: We hope we clarified this misunderstanding with the additionally added sections. We in fact assume local consistency of dynamics in local temporal segments as illustrated in Figure 5 in Appendix B by having a rather flexible definition for a task. We assume each task as modelling the locally consistent dynamics over these temporal segments and the latent task variable $l$ models a distribution over the tasks/dynamical systems. Yet our formalism can model the global changes in dynamics since we obtain a different instance of the HiP-RSSM for each temporal segment based on the observed context set.
>
> **Concern 3**: “Why are permutation invariant deep models used for encoding the history of past N steps? Why is it useful to discard the temporal information?”*
>
> **Reply**: The use of permutation invariant set encoders to capture distribution over functions (in our case transition functions in latent space) has been inspired by related works on modelling distribution over functions in Neural Process ( https://arxiv.org/abs/1807.01622 ) literature, where permutation invariance (exchangeability) in addition to consistency forms necessary and sufficient condition for modelling distribution over functions (stochastic processes). Recent works have shown how the set encoders in NPs can play the role of deep kernels in GPs (/http://bayesiandeeplearning.org/2018/papers/128.pdf) which models distribution over functions.
> We further empirically back our choice by conducting an ablation study where we compare the set encoder with a recurrent encoder in Appendix E. Processing using set encoders adds minimal additional computational/memory constraints since they allow parallelization in processing the context set as opposed to the recurrent encoders. Moreover, the additional complexity of the recurrent encoder does not result in better performance as seen in our abalations, hence, we opted for the set encoder.
>
> **Concern 4**: *“The model assumes that the hidden parameter value is fixed throughout and does not consider at all how the current hidden parameter influences the next value of the hidden parameter. Further, the paper misses out on recent works by Xie et al. (2020) that build upon the work by Nagabandi et al. (2018a;b) and Finn et al. (2017). They look at ..... they make the assumption of dynamics having a locally fixed hidden parameter like this work, they also consider how the hidden parameter is evolving more globally.”*
>
> **Reply**: We agree that further explicitly modelling the transition of these latent task variables may lead to better generalization and knowledge transfer. However, even with our simpler assumptions which considerably simplifies the procedure for inferring the hidden parametrization, we are still able to capture the global evolution of parameters rather efficiently as seen in figures 4b and 4c. The simplicity also makes the entire approach scalable to be deployed on real robotic applications.
>
> We would also like to point out that, as opposed to the recent publication Xie et al. (2020) which focus on solving model free RL tasks, we focus on recurrent state-space modelling / dynamics learning under multi-task scenarios. Nevertheless, the paper provides an interesting future direction to model the dynamics over latent task parameters which may result in better generalization and inter-task knowledge transfer.

---

> > ### Author Response · Authors · 2021-11-21
> > **Author Reply (Part 2)**
> >
> > **Concern 5**: “In section 3.2.2 It is mentioned that $z_{t-1}$, $l$,$a$ are independent of each other as $z_t$ is unobserved. How are $z_{t-1}$ and $l$ unobserved ? From figure 2 doesnot $l$ directly influence $z_{t-1}$ ?"
> >
> > **Reply**: Sorry, for the lack of clarity and referring to figure 2 may mislead the readers. We have clarified this in section 3.2.2. We use a forward algorithm to perform inference, which is a recursive procedure consisting of Kalman time update and measurement update in the temporal order. The contribution of $l$ on $z_{t-1}$ at time $t-1$ is integrated out in the Kalman time update step at $t-1$ when we compute the prior over latent states $z_{t-1}$. We then further calculate the posterior $(z_{t-1}|w_{1:t-1},a_{1:t-1},C_l)$ at $t-1$.
> >
> > Thus at any time t, it’s the posterior over the belief state $(z_{t-1}|w_{1:t-1},a_{1:t-1},C_l)$ , posterior over the latent task variable $l|C_{l}$ and the action $a_t$ are independent of each other since they form a ``common-effect" / v-structure with the unobserved variable $z_{t}$}.
> >
> > **Concern 6**: “In section 3.2.2 what is the random variable being sampled from the third Gaussian distribution? Is it supposed to be $y$ instead of $v$ ?"
> >
> > **Reply**:  Sorry for the confusion caused by the use of confusing notations. It is in fact sampled from a conditional distribution $y|u,v$ and not v. However we have used a clearer notation in the updated version to avoid misunderstandings.
> >
> > **Concern 7**: “Should $z_t^-$ depended on  $z_{t-1}^-$ instead of  $z_{t-1}$"
> >
> > **Reply**:  Thank you for pointing out this typo. It is the posterior latent state from $t-1$, $z_{t-1}^+$.
> >
> > **Concern 8**: “Is matrix A time-varying or fixed ?? Notations are used inter-changeably."
> >
> > **Reply**:  We use a time-varying locally linear transition model A in HiP-RSSM. We use the consistent notation $A_t$ now.
> >
> > We hope we could clarify/remove most of your concerns. We invite you to ask additional questions and engage in further discussion if this is not the case.

---

> > > ### Author Response · Authors · 2021-11-29
> > > **Request For Feedback**
> > >
> > > Dear Reviewer,
> > >
> > > Thanks again for your valuable comments and reviews. We hope that you've had a chance to read our response and updated version of the paper. We would really appreciate a reply as to whether our response and clarifications have addressed the issues raised in your review, or whether there is anything else we can address.
> > >
> > > Authors

---

> > > > ### Comment · Reviewer_VmRd · 2021-12-03
> > > > **Thank you for clarifications**
> > > >
> > > >
> > > > Thank you for all the clarifications. I have increased my score but still lean towards rejection. I encourage the authors to pursue their interesting work further but I believe that currently the work can benefit significantly by placing itself better in comparison to the related work and being more clear about the contributions.
> > > >
> > > > - Ignoring how new latent task parameters depend on the previous one and modeling it as multi-task setup sidelines the core of the non-stationary problem. Multi-task learning is essentially just a _stationary_ POMDP setting where the start state distribution is also used to draw the (hidden) task parameter. From this point of view, I think the existing deep SSM methods are immediately applicable.
> > > >
> > > > - Compared to the RNN baselines that do not need to assume fixed local structure, the proposed method needs to decide context length during training/inference. Since the duration of local stationarity is not known a-priori it is important to analyze the impact of such parameters. I guess this is an important drawback of ignoring how new latent task parameters depend on the previous one.
> > > >
> > > > - Quick skim of the switching dynamical systems works that other reviewers brought up suggests that they are indeed modeling how new latent task parameters depend on the previous one. Yes, they do consider discrete switches but is there any fundamental reason why their method cannot be directly extended by using Normal distributions?
> > > >
> > > > - Further, while it is true that Xie et al. (2020) focus on control, they are already modeling the non-stationary dynamics as a sub-problem in their algorithm, while also considering how previous latent variable influences the new one. Comparing against their method could be insightful.
> > > >
> > > >  I think these works warrant more thorough comparison, particularly not just quantitative but also qualitative.
> > > >
> > > >
> > > > - While the paper presents in the appendix all the hyper-parameters used finally, providing more details on how hyper-parameters were searched for the proposed methods and the baseline, and sensitivity to different hyper-parameters would be more informative.
> > > >
> > > > - Finally, there are a lot of assumptions in the proposed work (linear evolution, Gaussian noise, etc.) that deserves more discussion in terms of when should one consider these reasonable.

---

> > > > > ### Author Response · Authors · 2021-12-06
> > > > > **Thanks for the feedback**
> > > > >
> > > > > Many thanks for your reply and updated score. Thank you for your suggestions.
> > > > >
> > > > > Regarding your observation on Deep-SSMs, we would like to point out that existing literature in Deep SSMs (eg:  https://arxiv.org/abs/1710.05741, https://arxiv.org/abs/1605.06432 , https://arxiv.org/abs/1905.07357 etc) are variants for learning in the same graphical model with different inference schemes (variational, closed-form inference etc). All these models do not assume that dynamics can be changing neither in their modelling assumptions nor in their experiments. To account for this we introduce the formalism of HiP-SSM (instead of SSMs) with the same motivation as to why a HIP-MDP (instead of MDPs) was formalized. We say this formalism which can have closed-form updates, enable deep SSMs to handle changing dynamics scenarios as seen in our quantitative and qualitative experiments. The experimental section clearly shows how an existing Deep-SSM (https://arxiv.org/abs/1905.07357  ) fails in this case.
> > > > >
> > > > > We agree that a comparison with switching dynamical systems which is a different formalism for the same problem would be an important one. A quantitative/qualitative comparison could not be performed unfortunately because of time constraints. However, most of these models have been validated only on small datasets to the best of our knowledge and how these would scale to larger robotic settings with both discrete and continuous changes in dynamics as we do would be an important one to analyze.
> > > > >
> > > > > Thanks again for your valuable feedback.
> > > > >
> > > > > Authors

---

### Decision · Program_Chairs · 2022-01-20

**Decision:**

Accept (Poster)

**Comment:**

The paper addresses a few very important points on sequential latent-variable models, and introduce a different view on meta-RL.  Even  though the methods that this paper poses are incremental, it is such a hot-debated topic that I would prefer to see this published now.